



**Trade-offs between water loss and carbon gain in a subtropical**
**primary forest on Karst soils in China**
Jing Wang[1,2,3]; Xuefa Wen[1,2*], Xinyu Zhang[1,2*], Shenggong Li[1,2]
1 Key Laboratory of Ecosystem Network Observation and Modeling, Institute of Geographic
Sciences and Natural Resources Research, Chinese Academy of Sciences, Beijing 100101, China
2 College of Resources and Environment, University of Chinese Academy of Sciences, Beijing
100190, China
3 School of Life Sciences, Beijing Normal University, Beijing 100875, China
*Correspondence: Xuefa Wen (Email: wenxf@igsnrr.ac.cn. Phone +86-010-64889272)
and Xinyu Zhang (zhangxy@igsnrr.ac.cn. Phone +86-10-64889679)





## Abstract:

Little attention has been given to plants's trade-off between carbon gain and water loss in Karst Critical Zone in southwestern China with low soil nutrient and water availability. An advanced understanding of the impact of $CO_2$ diffusion and maximum carboxylase activity of Rubisco ($V_{cmax}$) on the light-saturated net photosynthesis ($A$) and intrinsic water use efficiency (iWUE) in Karst plants can provide insight into physiological strategies used in adaptation to harsh environments. We selected six plant life forms (63 species) in a subtropical Karst primary forest in southwestern China, and measured $CO_2$ response curves, and calculated corresponding stomatal conductance to $CO_2$ ($g_s$), mesophyll conductance to $CO_2$ ($g_m$), and $V_{cmax}$. The results showed that $g_s$ varied from 0.05 to 0.38 mol $CO_2$ m$^{-2}$ s$^{-1}$, $g_m$ varied from 0.02 to 0.69 mol $CO_2$ m$^{-2}$ s$^{-1}$, and $g_m$ was positively related to $g_s$; foliar $A$ was co-limited by $g_s$, $g_m$, and $V_{cmax}$ in trees, tree/shrubs, and shrubs with relatively high leaf mass per area (LMA), and mainly constrained by $g_m$ in grasses, vines, and ferns with relatively low LMA; and iWUE varied from 29.52 to 88.92 µmol $CO_2$ mol$^{-1}$ $H_2O$ across all species, and was significantly correlated with $g_s$, $g_m/g_s$, and $V_{cmax}/g_s$. These results indicated that Karst plants maintained relatively high $A$ and low iWUE through the co-variation of $g_s$, $g_m$, and $V_{cmax}$ as adaptation to Karst environment.

**Key words:** iWUE; mesophyll conductance; stomatal conductance; Karst critical zone; $V_{cmax}$



## 1 Introduction

The Karst Critical Zone (Karst CZ) in southwestern China accounts for over 12% of the total global land area (more than $54 \times 10^4$ km$^2$) (Zhang et al., 2011). Compared with other CZs developed on other lithologies, Karst CZ was developed on limestone bedrock, and characterized by inhomogeneous and shallow soil due to the greater hydraulic erosion and complex underground drainage network (Nie et al., 2014; Chen et al., 2015). In such conditions, the soil cannot retain enough nutrients and water for plant growth even though precipitation is high (1000-2000 mm) (Liu et al., 2011; Fu et al., 2012; Chen et al., 2015). To adapt to the harsh environment, Karst plants develop distinct patterns of light-saturated net photosynthesis ($A$) and trade-off between carbon gain and water loss to adapt to the harsh environment (Sullivan et al., 2017). The intrinsic water use efficiency (iWUE=$A/g_{sw}$, the ratio of $A$ to stomatal conductance to H$_2$O ($g_{sw}$)), is an effective indicator of the trade-off between carbon gain and water loss (Moreno-Gutierrez et al., 2012). Until now, variability in $A$ and iWUE has been reported only in 13 co-occurring trees and 12 vines (Chen et al., 2015), and 12 co-occurring tree species (Fu et al., 2012) in two tropical Karst forests in southwestern China.

Based on Fick's first law, $A$ has been shown to be limited only by leaf stomatal conductance to CO$_2$ ($g_s = g_{sw}/1.6$) and $V_{cmax}$ (Flexas et al., 2012; Buckley and Warren, 2014); originally, mesophyll conductance to CO$_2$ ($g_m$) was proposed to be infinite, i.e. CO$_2$ concentration in chloroplast ($C_c$) was equal to the CO$_2$ concentration in intercellular air space ($C_i$). Indeed, $g_m$ varies greatly among species (Warren and Adams, 2006; Flexas et al., 2013). Recent studies have confirmed that $A$ was constrained jointly by $g_s$, $g_m$, and $V_{cmax}$, and their relative contribution to $A$ was species-dependent and site-specific (Carriqui et al., 2015; Tosens et al., 2016; Galmes et al., 2017; Peguero-Pina et al., 2017a; Peguero-Pina et al., 2017b; Veromann-Jurgenson et al., 2017).



Variation in iWUE ($=A/g_{sw}$) depends on the relative changes in $A$ ($g_s$, $g_m$, $V_{cmax}$) and
$g_{sw}$ ($g_{sw}=1.6g_s$) (Flexas et al., 2013; Gago et al., 2014). Theoretical relationships
between iWUE and $g_s$, $g_m$, and $V_{cmax}$ have been deduced using two approaches. Based
on Fick's first law of $CO_2$ diffusion, Flexas et al. (2013) deduced that iWUE was a
function of $g_m/g_s$ and $CO_2$ gradients ($C_a$-$C_c$) within leaf. On the other hand, combining
Fick's first law of $CO_2$ diffusion and Farquhar biochemical model (Farquhar and
Sharkey, 1982), Flexas et al. (2016) deduced that iWUE was a function of $V_{cmax}/g_s$, $C_c$,
$CO_2$ compensation point of photosynthesis ($\Gamma^*$), and the effective Michaelis–Menten
constant of Rubisco for $CO_2$ ($K_m$). Until now, most previous studies focused on the
role of $CO_2$ diffusion in limiting iWUE, and suggested that iWUE was negatively
related to $g_s$, and positively related to $g_m/g_s$ (Flexas et al., 2013). Gago et al. (2014)
used a meta-analysis with 239 species, and were the first to confirm that iWUE was
positively related to $V_{cmax}/g_s$. Although both $g_m/g_s$ and $V_{cmax}/g_s$ were positively
correlated with iWUE, there was only a weak correlation between $g_m/g_s$ and $V_{cmax}/g_s$,
which indicates that iWUE can be improved by increasing $V_{cmax}$ or $g_m$ (proportionally
higher than $g_s$), not both (Gago et al., 2014).

It is noteworthy that Flexas et al. (2016) and Gago et al. (2014) found that most of the
previous work on constraints of $g_s$, $g_m$, and $V_{cmax}$ on $A$ were conducted in crops or
saplings, and only a few studies were in natural ecosystems. For example, $g_m$ was the
main factor limiting $A$ in two Antarctic vascular grasses (Saez et al., 2017), and in 35
Australian sclerophylls (Niinemets et al., 2009b) in different habitats. The $A$ of two
closely-related Mediterranean *Abies* species growing in two different habitats was
mainly constrained by $g_m$ in one, and by $g_s$ in the other habitat (Peguero-Pina et al.,
2012). Beyond that, it still remains unknown how $g_s$, $g_m$, and $V_{cmax}$ regulate $A$ and
iWUE across species in natural ecosystems.

In this study, we selected 63 dominant plant species, including six life forms (29 trees,
11 trees/shrubs, 11 shrubs, 4 grasses, 5 vines, and 3 ferns), from a subtropical primary
forest in the Karst CZ of southwestern China, and measured their $A$ and $CO_2$ response



curves. The $g_m$ was calculated using the curve-fitting method (Ethier and Livingston,
2004). The obtained $g_m$ was used to transform the $A$-$C_i$ into $A$-$C_c$ response curves, and
then to calculate the $A$ and $V_{cmax}$. Our objective was to determine and distinguish the
limitations of $CO_2$ diffusion ($g_s$ and $g_m$) and $V_{cmax}$ on $A$ and iWUE in different life
forms in this Karst primary forest.

## 2 Materials and Methods

### 2.1 Site information

This study was conducted in a subtropical primary forest (26°14′48″N, 105°45′51″E;
elevation, 1460 m), located in the Karst CZ of southwestern China. This region has a
typical subtropical monsoon climate, with a mean annual precipitation of 1255 mm,
and mean annual air temperature of 15.1 °C (Zeng et al., 2016). The soils are
characterized by a high ratio of exposed rock, shallow and nonhomogeneous soil
cover, and complex underground drainage networks, e.g. grooves, channels and
depressions (Chen et al., 2010; Zhang et al., 2011; Wen et al., 2016). Soils and soil
water are easily leached into underground drainage networks. Soil texture was
silt-clay loam, and soil PH was 6.80±0.16 (Chang et al., 2018). The total nitrogen
and phosphorus content in soil was 7.30±0.66 and 1.18±0.35 g $Kg^{-1}$, respectively,
which was similar with that of non-Karst CZs (Wang et al., in review). However, the
soil quantities (16.04~61.89 Kg $m^{-2}$) and nitrogen and phosphorus storage (12.04 and
1.68 t $hm^{-2}$) was much lower than that of non-Karst CZs, due to the thin and
heterogeneous soil layer (He et al., 2008; Jobbagy et al., 2000; Lu et al., 2010; Li et
al., 2008). The typical vegetation type is mixed evergreen and broadleaf deciduous
primary forest, dominated by *Itea yunnanensis* Franch, *Carpinus pubescens* Burk*.,*
and *Lithocarpus confinis* Huang, etc. (Wang et al., in review).

### 2.2 Leaf gas-exchange measurements

In July and August 2016, 63 dominant species of six life forms (Table S1), including
29 trees, 11 trees/shrubs, 4 shrubs, 4 grasses, 5 vines, and 3 ferns, were selected for





measurements of the $A$ and $CO_2$ response curves. Details of leaf sampling and
measurements of $CO_2$ response-curves were described in Wang et al. (in review).
Briefly, a total of 189 fully sun-exposed, mature leaves were collected from adult
individuals of 63 species to measure $CO_2$ response curves following procedural
guidelines (Longand Bernacchi, 2003) using a portable photosynthesis system
(Li-6400, Li-Cor, USA).

$A$ and the corresponding $g_{sw}$ ($g_s=g_{sw}/1.6$), $C_a$, and $C_i$ were extracted from the $CO_2$
response curve under saturating light (1500 μmol m$^{-2}$ s$^{-1}$) conditions, with $CO_2$
concentration inside the cuvette set to 400 μmol mol$^{-1}$ (Domingues et al., 2010;
Domingues et al., 2010). $V_{cmax}$ was estimated by fitting $A$-$C_c$ curves (Ethier and
Livingston, 2004). The obtained values of $g_m$ were used to transform the $A$-$C_i$ into
$A$-$C_c$ response curves as $C_c=C_i - A/g_m$.

The $g_m$ was calculated using a curve-fitting method (Ethier and Livingston, 2004). In
this study, calculated $C_c$ and the initial slope of $A$-$C_c$ curves were above zero,
indicating that $g_m$ estimated by the curve fitting method was valid (Warren and
Adams, 2006). Further details on the method to calculate $g_m$ are given in Section 4.1.

**2.3 Theory of trade-off between carbon and water at leaf scale**
The exchange of $H_2O$ and $CO_2$ between the leaf and the atmosphere is regulated by
stomata (Gago et al., 2014). According to Fick's first law of diffusion, $A$ and stomatal
conductance to $CO_2$ ($g_s$) are related as:

162        $A = g_s(C_a$-$C_i)$                          (1)

where $A$ is the photosynthetic rate (μmol $CO_2$ m$^{-2}$ s$^{-1}$); $C_a$ is the ambient $CO_2$
concentration (μmol mol$^{-1}$); $C_i$ is the intercellular $CO_2$ concentration (μmol mol$^{-1}$).

Besides stomata, mesophyll is another barrier for $CO_2$ inside the leaf. $A$ and
mesophyll conductance to $CO_2$ ($g_m$) are related as:





$A = g_m(C_i - C_c)$    (2)
where $C_c$ is the $CO_2$ concentration at the sites of carboxylation (μmol mol$^{-1}$). $C_c$ not
only depends on $CO_2$ supply by $g_m$, but also on $CO_2$ demand (the maximum
carboxylase activity of Rubisco, $V_{cmax}$).

**(1) The relationship between iWUE and $g_m/g_s$**
iWUE is a function of $CO_2$ diffusion conductances (e.g. $g_s$ and $g_m$) and leaf $CO_2$
concentration gradients. We can express $A$ as the product of the total $CO_2$ diffusion
conductance ($g_t$) from ambient air to chloroplasts, and the corresponding $CO_2$
concentration gradients by combining Eq. (1) and (2) (Flexas et al., 2013):
$A = g_t \left[ (C_a - C_i) + (C_i - C_c) \right]$    (3)
where $g_t = 1/(1/g_s + 1/g_m)$. This equation demonstrates that $CO_2$ concentration gradients
in leaves are constrained by stomatal and mesophyll resistance to $CO_2$. Therefore,
iWUE can be expressed as:
$$\frac{A}{g_{sw}} = \frac{1}{1.6} \left( \frac{g_m/g_s}{1 + g_m/g_s} \right) \left[ (C_a - C_i) + (C_i - C_c) \right]$$

(4)

Eq. (4) means that iWUE is positively related to $g_m/g_s$, but not to $g_m$ itself (Warren
and Adams, 2006; Flexas et al., 2013; Buckley and Warren, 2014; Cano et al., 2014).

**(2) The relationship between iWUE and $V_{cmax}/g_s$**
When Fick's first law and the Farquhar biochemical model (Farquhar and Sharkey,
1982) are combined, iWUE is also a function of $V_{cmax}$. Based on the Farquhar
biochemical model (Farquhar and Sharkey, 1982), when $A$ is limited by Rubisco, it
can be expressed by the following equation (Sharkey et al., 2007):
$A = \dfrac{V_{cmax}(C_c - \Gamma^*)}{(C_c + K_m)} - R_d$    (5)

where $\Gamma^*$ is the $CO_2$ compensation point of photosynthesis in the absence of
non-photorespiratory respiration in light ($R_d$), and $K_m$ is the effective
Michaelis–Menten constant of Rubisco for $CO_2$. Combining Eq. (1) and (5) (Flexas et
al., 2016), we obtain:

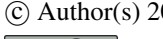




$$\frac{V_{cmax}}{g_s} = \frac{(C_c+K_m)(C_a-C_i)(A+R_d)}{(C_c-\Gamma^*)A}$$

(6)

Because $R_d$ is much smaller than $A$ in actively photosynthesizing leaves, $V_{cmax}/g_s$ can
be approximated as:

$$\frac{V_{cmax}}{g_s} \approx \frac{(C_c+K_m)(C_a-C_i)}{(C_c-\Gamma^*)} = \frac{(C_c+K_m)}{(C_c-\Gamma^*)}\frac{A}{g_s}$$

(7)

Consequently, iWUE can be expressed as:

$$\frac{A}{g_{sw}} = \frac{1}{1.6}\frac{V_{cmax}}{g_s}\frac{(C_c-\Gamma^*)}{(C_c+K_m)}$$

(8)


## 208 2.4 Quantitative analysis of limitations on $A$

The relative contribution of $g_s$ ($l_s$), $g_m$ ($l_m$) and $V_{cmax}$ ($l_b$) to $A$ can be separated by a
quantitative limitation model introduced by Jones (Jones, 1985) and further developed
by Grassi & Magnani (2005). The sum of $l_s$, $l_m$, and $l_b$ is 1. $l_s$, $l_m$ and $l_b$ can be
calculated as:


$$l_s = \frac{g_t/g_s \cdot \partial A/\partial C_c}{g_t+\partial A/\partial C_c}$$

(12)



$$l_m = \frac{g_t/g_m \cdot \partial A/\partial C_c}{g_t+\partial A/\partial C_c}$$

(13)



$$l_b = \frac{g_t}{g_t+\partial A/\partial C_c}$$

(14)


where $\partial A/\partial C_c$ was calculated as the slope of $A$-$C_c$ response curves over a $C_c$ range of
50–100 μmol mol$^{-1}$. $l_s$, $l_m$ and $l_b$ have no units. $A$ is co-limited by the three factors
when $l_s \approx 0.3$, $l_m \approx 0.3$ and $l_b \approx 0.4$ (Galmes, J. et al., 2017).

## 224 2.5 Statistical analysis

The correlation analysis was performed using the least square method, and all of the
data were log$_e$-transformed. The probability of significance was defined at p< 0.05.

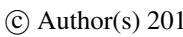




## 3 Results

### 3.1 Interrelation among $g_s$, $g_m$, $g_t$, and $V_{cmax}$

$CO_2$ concentration gradients in leaf were controlled by $CO_2$ diffusion conductance
and $V_{cmax}$. Fig. 1 shows the relationship between $CO_2$ gradients ($C_a$-$C_i$, $C_i$-$C_c$ and
$C_a$-$C_c$) in leaf and the corresponding $CO_2$ diffusion conductance ($g_s$, $g_m$ and $g_t$) (Fig.
1a-c), and between $C_a$-$C_c$ and $V_{cmax}$ (Fig. 1d). $CO_2$ concentration gradients ($C_a$-$C_i$,
$C_i$-$C_c$ and $C_a$-$C_c$) were significantly negatively associated with the corresponding $CO_2$
diffusion conductance ($g_s$, $g_m$ and $g_t$) ($P<0.001$). $V_{cmax}$ was positively associated with
$C_a$-$C_c$ ($P<0.001$).

The $g_s$, $g_m$, and $g_t$ were significantly positively related to each other ($P<0.001$) (Fig.
S1). The contribution of $g_m$ to leaf $CO_2$ gradient was similar to that of $g_s$ (Fig. S3).
The contribution of $g_s$ (57.51–155.13 µmol mol$^{-1}$) to $C_a$-$C_c$ (98.50–282.94 µmol mol$^{-1}$)
varied from 28% to 86%, and the contribution of $g_m$ (18.15–179.36 µmol mol$^{-1}$) to
$C_a$-$C_c$ varied from 14% to 72%. But the variation range of $g_m$ (0.02 –0.69 mol $CO_2$ m$^{-2}$
s$^{-1}$) was 4.5 times that of $g_s$ (0.05–0.38 mol $CO_2$ m$^{-2}$ s$^{-1}$).

No relationship was found between the $CO_2$ diffusion conductance ($g_s$, $g_m$, and $g_t$) and
$V_{cmax}$ (Fig. S2). However, after normalization of $g_s$, $g_m$, $g_t$, and $V_{cmax}$ for $A$ (normalized
parameters are hereafter called $G_S=g_S/A$, $G_m=g_m/A$, $G_t=g_t/A$, and $V=V_{cmax}/A$), $V$ was
significantly positively correlated with $G_m$ and $G_t$ ($P<0.001$) (Fig. 2b and c), and was
slightly positively correlated with $G_s$ ($P<0.05$) (Fig. 2a), which represented the
trade-off between $CO_2$ supply and demand.

### 3.2 Contribution of $g_s$, $g_m$ and $V_{cmax}$ to A

The variation in $A$ was attributed to variation in both of $g_s$, $g_m$, $g_t$, and $V_{cmax}$. $A$ was
positively correlated with $g_s$ (Fig. 3a), $g_m$ (Fig. 3b), and $V_{cmax}$ (Fig. 3c). We used the



quantitative limitation model (Eqs. (13), (14) and (15)) to separate contributions by $g_s$
($l_s$), $g_m$ ($l_m$), and $V_{cmax}$ ($l_b$) to limiting $A$. The $l_s$, $l_m$, and $l_b$ were negatively associated
with, respectively, $g_s$, $g_m$, and $V_{cmax}$ (Fig. 4). The contributions by $g_s$, $g_m$, and $V_{cmax}$ to
limiting $A$ were different for each species (Fig. S3). $l_s$ varied from 0.17 to 0.45
(2.6-fold), $l_m$ varied from 0.05 to 0.55 (10.5-fold), and $l_b$ varied from 0.11 to 0.68
(6.2-fold) across species. Overall, $g_m$ contribution to limiting $A$ was the largest
($l_m$=0.38±0.12), followed by $V_{cmax}$ ($l_b$=0.34±0.11), and $g_s$ ($l_s$=0.28±0.07).

To further understand how $A$ was limited by $g_s$, $g_m$, and $V_{cmax}$, we grouped the 63
species into 6 life forms: tree, tree/shrub, shrub, grass, vine, and fern. The averaged
leaf mass per area (LMA) of the 6 life forms above were as follows: 69.41±29.31,
93.94±27.89, 72.35±42.37, 47.08±16.39, 40.86±13.22 and 44.21±12.35 g m$^{-2}$.
The results showed that tree, tree/shrub, and shrub with relatively high LMA were
co-limited by $g_s$, $g_m$, and $V_{cmax}$, while $g_m$ was the main constrain factor for the other
three life forms with relatively low LMA (Fig. 5). The $l_s$ showed a decreasing trend
from tree to fern. The largest average value of $l_s$ was observed for tree and tree/shrub,
followed by shrub, grass, and vine and fern. The $l_m$ first declined, and then increased.
Grass had the largest averaged value of $l_m$. In contrast, $l_b$ first increased and then
decreased. Grass had the smallest averaged value of $l_b$.

## 3.3 Effect of $g_s$, $g_m$ and $V_{cmax}$ on iWUE

The iWUE varied from 29.52 to 88.92 μmol $CO_2$ mol$^{-1}$ $H_2O$. In theory, iWUE is
regulated by $g_s$ ($g_{sw}$=1.6$g_s$), $g_m$, and $V_{cmax}$. However, a simple correlation analysis
showed that iWUE was negatively related to $g_s$ (Fig. 6b), and not related to $A$ (Fig.
6a), $g_m$ (Fig. 6c), and $V_{cmax}$ (Fig. 6d).

A correlation analysis was used to test how $g_m/g_s$ and $V_{cmax}/g_s$ affected iWUE. The
results showed that iWUE was positively correlated with $g_m/g_s$ (Fig. 7a) and $V_{cmax}/g_s$
(Fig. 7b). However, there was no significant relationship between $g_m/g_s$ and $V_{cmax}/g_s$.





The iWUE was regulated by co-variation between $g_s$, $g_m$ and $V_{cmax}$.

## 4 Discussion

### 4.1 The role of $g_m$ in CO$_2$ diffusion and $V_{cmax}$

Three methods are most commonly used for $g_m$ estimation. Those methods have been
reviewed by Warren (2006) and Pons et al. (2009). Briefly, $g_m$ can be calculated by
the stable isotope method (Evans, 1983; Sharkey et al., 1991; Loreto et al., 1992), $J$
method (Bongi and Loreto, 1989; Dimarco et al., 1990; Harley et al., 1992; Epron et
al., 1995; Laisk et al., 2005), and 'curve-fitting' method (Ethier and Livingston, 2004;
Sharkey et al., 2007). All of these methods are based on gas exchange measurements
(Pons et al., 2009), and some common assumptions (Warren, 2006). Thus, the
accuracy of each method is largely unknown (Warren, 2006).

The $g_m$ was estimated by the 'curve-fitting' method in this study. Although the
'curve-fitting' method is less precise than the stable isotope method, the
'curve-fitting' method is much more readily available and has been used for several
decades (Warren, 2006; Sharkey, 2012). Accurate measurements of $A$ and $C_i$ is a
prerequisite for estimating $g_m$ using the 'curve-fitting' method (Pons et al., 2009).
Warren (2006) pointed out that highly-accurate measurements need small leaf area
and low flow rates. We confirmed that the calculated $C_c$ and the initial slope of $A$-$C_c$
curves were positive, suggesting that the measured $g_m$ was reliable (Warren, 2006).

Large variability in $g_m$ has been shown both between and within species with different
leaf forms and habits (Gago et al., 2014; Flexas et al., 2016). Variability in $g_m$ in this
study is similar to that in global datasets (Gago et al., 2014; Flexas et al., 2016). The
order of averaged $g_m$ from different life forms was as follows: tree > tree/shrub >
grass > shrub > vine > fern. Previous studies have confirmed that the liquid phase of
mesophyll (Veromann-Jurgenson et al., 2017), cell wall thickness of mesophyll
(Terashima et al., 2011) or chloroplast (Tosens et al., 2016), and surface area of




mesophyll and chloroplast exposed to intercellular space (Veromann-Jurgenson et al.,
2017) were the main limitations for $g_m$. The LMA varied from 22.98 g m$^{-2}$ to 154.61 g
m$^{-2}$, the averaged value was 69.32$\pm$32.70 g m$^{-2}$ (Wang et al., in review). Hence, the
wide variability of $g_m$ between different species and life forms in the same ecosystem
seems to be related to the diversity in leaf anatomical traits.

Large uncertainties can be introduced by ignoring $g_m$. On one hand, $g_m$ plays a similar
or somewhat lesser role than $g_s$ in $CO_2$ diffusion in leaf (Warren, 2006). In the present
study, $g_m$ was positively related to $g_s$ (Fig. S1), variability range of $g_m$ was larger than
that of $g_s$, and the contribution of $g_m$ to $C_a$-$C_c$ was similar to that of $g_s$. Hence,
ignoring $g_m$ would overestimate the carbon isotope discrimination in photosynthesis
($\Delta^{13}$C) (von-Caemmerer, 1996; Warren, 2006). Consistent with previous studies
(von-Caemmerer, 1996; Warren, 2006), there was a significantly positive relationship
between $\Delta^{13}$C_$g_m$ and $\Delta^{13}$C_$g_s$ ($\Delta^{13}$C_$g_m$=2.38*$\Delta^{13}$C_$g_s$-35.54, R$^2$=0.22, P<0.001).
$\Delta^{13}$C_$g_m$ represented the carbon isotope discrimination when $g_m$ was finite, and
$\Delta^{13}$C_$g_s$ represented the carbon isotope discrimination when $g_m$ was infinite.

On the other hand, ignoring $g_m$ would underestimate $V_{cmax}$ up to 75% (Sun et al.,
2014). In this study, the relationship between V$_{cmax\_Ci}$ and V$_{cmax\_Cc}$ can be expressed as:
$V_{cmax\_Cc}$=2.6*$V_{cmax\_Cc}$-22.12 ($R^2$=0.25, $P$<0.001). $V_{cmax\_Ci}$ represented $V_{cmax}$ calculated
based on the $A$-$C_i$ curve, and $V_{cmax\_Cc}$ represented $V_{cmax}$ calculated based on the $A$-$C_c$
curve. Furthermore, the leaf barrier to $CO_2$ caused by $g_m$ has not been represented in
the global carbon cycles, leading to an overestimation of $CO_2$ supply for carboxylation
and an underestimation of the response of photosynthesis to atmospheric $CO_2$ (Sun et
al., 2014).

**4.2 Co-variation in $g_s$, $g_m$ and $V_{cmax}$ in regulating $A$**
The $A$ was constrained by $g_s$, $g_m$, and $V_{cmax}$ acting together, however, variability in the
relative contribution of these three factors depended on species and habitats (Tosens
et al., 2016; Galmes et al., 2017; Peguero-Pina et al., 2017a; Veromann-Jurgenson et





al., 2017). Compared with the global dataset, the $A$ in the study site was high at a
given leaf phosphorus (P) level (Wang et al., in review). Under well-watered
conditions, $A$ was co-limited by the three factors in angiosperm species (Galmes et al.,
2017), and mainly limited by $g_m$ in ferns (Carriqui et al., 2015). Similarly in the
present study, $A$ of tree, tree/shrub, and shrub was co-limited by $g_s$, $g_m$, and $V_{cmax}$, and
$A$ of fern was mainly limited by $g_m$. However, $A$ of both grass and vine was mainly
limited by $g_m$ (average $l_m$>0.4, with the largest value of 0.55 and 0.54 for grass and
fern, respectively). In addition, 20 of the 63 species were mainly limited by $V_{cmax}$
($l_b$>0.4, with the largest value of 0.68).

The importance of $g_s$ and $g_m$ in constraining $A$ was variable, and depended on leaf and
mesophyll structural traits, i.e. LMA (Tomas et al., 2013), and thickness of leaf, cell
wall (Peguero-Pina et al., 2017b), and mesophyll itself (Giuliani et al., 2013). The
negative correlation of $g_m$ with LMA has been reported in previous studies (Niinemets
et al., 2009a; Tomas et al., 2013). The lack of correlation between $g_m$ and LMA, and a
positive relationship between $g_m$/LMA and LMA in this study were similar to those
shown for gymnosperms (Veromann-Jurgenson et al., 2017). The reason for the
similarities may be a strong investment in supportive structures (Veromann-Jurgenson
et al., 2017).

$A$ of species with low LMA was co-limited by $g_s$, $g_m$, and $V_{cmax}$, while $A$ of species
with high LMA was mainly limited by $CO_2$ diffusion (Tomas et al., 2013). In this
study, trees, tree/shrub, and shrubs with relatively high LMA were co-limited by $g_s$,
$g_m$, and $V_{cmax}$, and life forms with low LMA were mainly limited by $g_m$. Furthermore,
we found that $g_m$ was positively related to $A$ ($R^2$=0.54, $P$<0.001, Fig. 3b), however,
there was no close relationship between $g_m$ and LMA. The reason for this may be that
species with high LMA may have thin cell walls in mesophyll (Terashima et al.,
2011), and chloroplast (Tosens et al., 2016), or large surface areas of mesophyll and
chloroplast exposed to intercellular space (Veromann-Jurgenson et al., 2017);
conversely, species with low LMA may have thin cell walls in mesophyll (Terashima



et al., 2011), and chloroplast (Tosens et al., 2016), or small surface areas of mesophyll
and chloroplast exposed to intercellular space (Veromann-Jurgenson et al., 2017).

Furthermore, the co-variation of $g_s$ and $g_m$ can also regulate $A$. Both $g_s$ and $g_m$ are
important physical determinants of $CO_2$ supply from the atmosphere to the
chloroplasts (Giuliani et al., 2013). The restricted $CO_2$ diffusion from the ambient air
to chloroplast is the main reason for a decreased $A$ under water stress conditions due
to both the stomatal and mesophyll limitations (Olsovska et al., 2016). The
relationship between $g_s$ and $g_m$ may reflect a co-variation between $A$ and $g_m$, or a
tendency for $g_m$ to compensate for reductions in $g_s$ (Buckley and Warren, 2014).

The relative contribution of $V_{cmax}$ to $A$ not only depends on $C_a$-$C_c$, but also on leaf
nutrient levels. Leaf nitrogen (N) and P were closely related to $V_{cmax}$. Leaf N:P ratio in
the same plants in a related study was 24.55 ±7.7 (Wang et al., in review), indicating a
P limitation to photosynthesis (Gusewell 2004). Although there was no significant
relationship between $l_m$ and leaf N:P, there was a trend of increasing $l_m$ with
increasing leaf N:P.

The trade-off between $CO_2$ supply ($g_s$ and $g_m$) and demand (carboxylation capacity of
Rubisco) can help maintain high photosynthetic efficiency with low $CO_2$ diffusion
conductance (Galmes et al., 2017; Saez et al., 2017). In this study, we used $V_{cmax}$ as a
proxy for the carboxylation capacity of Rubisco, and the normalized $V_{cmax}$ by A
($V=V_{cmax}/A$) was significantly negatively correlated with the normalized $g_s$ by A ($G_t$
$=g_s/A$) (P<0.001) (Fig. 2c), indicating that the trade-off between $CO_2$ supply and
demand also existed among different species in the same ecosystems. For genus
*Limonium* (flowering plants) (Galmes et al., 2017), $g_t$ was significantly positively
related to Rubisco carboxylase specific activity, and significantly negatively related to
Rubisco specificity factor to $CO_2$. In case of Antarctic vascular (Saez et al., 2017) and
Mediterranean plants (Flexas et al., 2014), $A$ was mainly limited by low $g_m$, but it
could be partially counterbalanced by a highly-efficient Rubisco through high



specificity for $CO_2$. This highlights the importance of the trade-off between $CO_2$
supply and demand in plant adaptation to Karst environment. However, it is still
unknown how leaf anatomical traits affect $g_m$ and $A$, and this should be further
explored.

### 4.3 Co-variation of $g_s$, $g_m$ and $V_{cmax}$ in regulating iWUE

Compared with the global dataset under well-watered conditions (19.27-171.88 μmol
$CO_2$ mol$^{-1}$ $H_2O$) (Flexas et al., 2016), the iWUE (29.52-88.92 μmol $CO_2$ mol$^{-1}$ $H_2O$)
in this study was somewhat lower in this study. Although Karst soils cannot contain
enough water for plant growth, the water use strategies (high $g_s/A$ and low $V_{cmax\_Ci}/A$)
were similar to the shown for plants growing in hot and wet regions. Prentice et al.
(2014) studied the carbon gain and water loss of woody species in contrasting
climates, and found that species in hot and wet regions tend to loss more water in
order to fix more carbon (high $g_s/A$, low $V_{cmax\_Ci}/A$), and vice versa. These results
indicates that plants tend to loss more water in order to fix more carbon. However, the
variability of iWUE in this study was larger than in the Karst tropical primary forest
(Fu et al., 2012; Chen et al., 2015). The average iWUE of 12 vines and 13 trees in the
Karst tropical primary forest was 41.23±13.21 μmol $CO_2$ mol$^{-1}$ $H_2O$ (Chen et al.,
2015), while that of 6 evergreen and 6 deciduous trees was 66.7±4.9 and 49.7±2.0
μmol $CO_2$ mol$^{-1}$ $H_2O$, respectively (Fu et al., 2012)

The iWUE was regulated by the co-variation of $g_s$, $g_m$, and $V_{cmax}$. In theory, water loss
was regulated by $g_s$ only, while $A$ was regulated by $g_s$, $g_m$, and $V_{cmax}$ (Fig. 3) (Lawson
and and Blatt, 2014). However, iWUE in this study was negatively related to $g_s$, and
not related to $A$, $g_m$, or $V_{cmax}$ (Fig. 6). The reason for these relationships maybe that $A$,
$g_m$, and $V_{cmax}$ co-varied. First, $g_s$ was positively correlated to $g_m$. Second, an increase
in $V_{cmax}$ would inevitably reduce $C_c$ at a given $g_s$ and $g_m$ (Flexas et al., 2016). While
no significant relationship was found between $V_{cmax}$ and $CO_2$ diffusion conductance
($g_s$, $g_m$, and $g_t$), $V$ was negatively correlated with $G_s$, $G_m$, and $G_t$.






$CO_2$ diffusion and Farquhar biochemical model indicated that iWUE was affected by
$g_m/g_s$ and $V_{cmax}/g_s$ (Gago et al., 2014; Flexas et al., 2016). There was a hyperbolic
dependency of iWUE on $g_m/g_s$ due to the roles of $g_s$ and $g_m$ in $C_i$ and $C_c$, and of $C_c$ in
$A$ (Flexas et al., 2016). In meta-analyses, both Gago et al. (2014) and Flexas et al.
(2016) found that iWUE was significantly positively related to $g_m/g_s$ and $V_{cmax}/g_s$. The
results of this study are consistent with the meta-analyses (Fig. 7), demonstrating that
plant types with relatively high $g_m/g_s$ or $V_{cmax}/g_s$ had relatively high iWUE.

However, plants cannot simultaneously have high $g_m/g_s$ and high $V_{cmax}/g_s$. Similarly to
the study of Gago et al. (2014), we found no relationship between $g_m/g_s$ and $V_{cmax}/g_s$.
Gago et al. (2014) thought that the poor relationship between $g_m/g_s$ and $V_{cmax}/g_s$
indicated that the iWUE may be improved by $g_m/g_s$ or $V_{cmax}/g_s$ separately; if both of
them were simultaneously improved, the enhanced effect on iWUE could be
anticipated. In addition, Flexas et al. (2016) showed in a simulation that the increase
in iWUE caused by overinvestment in photosynthetic capacity would progressively
lead to inefficiency in trade-off between carbon gain and water use, causing an
imbalance between $CO_2$ supply and demand.

Water use strategies are critical to the survival and distribution of species, especially
in harsh environments, e.g. in low-nutrient availability and water stress (Nie et al.,
2014). Species with high $g_s$, and low iWUE were defined to have
'profligate/opportunistic' water use strategy, and species with low $g_s$ and high iWUE
were defined to exhibit 'conservative' water use strategy (Moreno-Gutierrez et al.,
2012). Species in Karst environment tended to lose more water to gain more carbon,
i.e. Karst plants using 'profligate/opportunistic' water use strategy to adapt to the
harsh enviroment,.

**5 Conclusions**



Our results studied the impact factors ($g_s$, $g_m$, and $V_{cmax}$) on $A$ and $iWUE$ in plants with
different life forms in field. The different contributions of $g_s$, $g_m$, and $V_{cmax}$ to $A$
indicated that plants used diverse trade-off between $CO_2$ supply and demand to
maintain relatively high $A$. iWUE was relatively low, but ranged widely, indicating
that plants used 'profligate/opportunistic' water use strategy to maintain the survival,
growth, and structure of the community. Those findings highlight the importance of
co-variation of $g_s$, $g_m$, and $V_{cmax}$ for the adaptation of plants to the harsh environment.
However, the effects of leaf anatomical traits on $g_s$, $g_m$, and the trade-off between leaf
anatomical traits and $V_{cmax}$ should be further explored.

## Acknowledgements

This study was supported by the National Natural Science Foundation of China
[41571130043, 31470500, and 41671257].

## Author contributions

JW, XFW. and XYZ planed and designed the research. JW performed experiments
and analyzed data. JW prepared the manuscript with contributions from all
co-authors.

## Competing interests.

The authors declare that they have no conflict of interest.

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





**Figures**

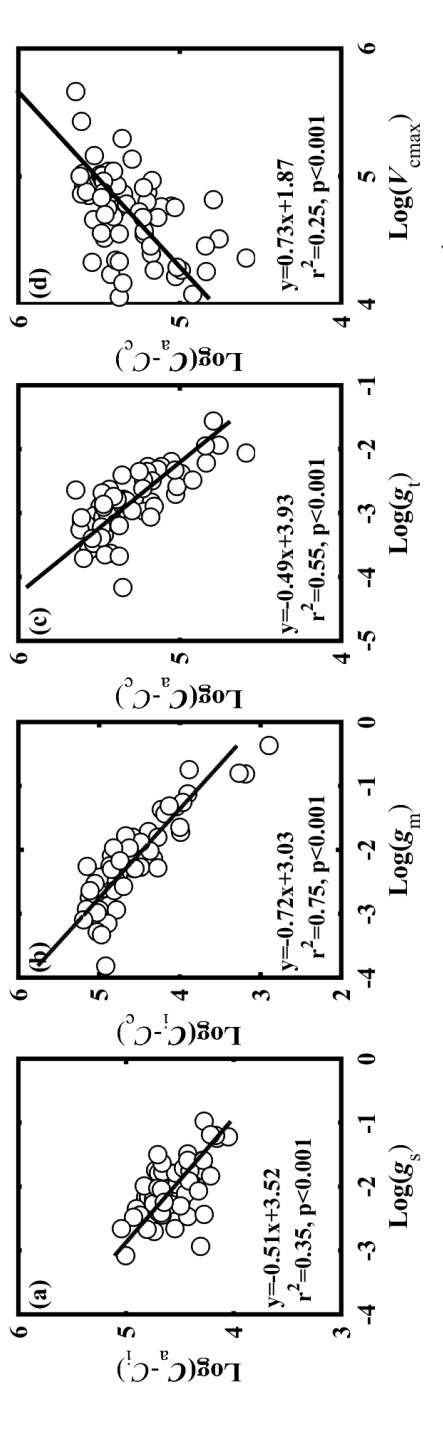


Figure 1. Relationships between (a) $CO_2$ gradient between ambient air and intercellular air space ($C_a$-$C_i$, µmol mol⁻¹) and stomatal conductance
to $CO_2$ ($g_s$, mol $CO_2$ m⁻² s⁻¹); (b) $CO_2$ gradient between intercellular air space and chloroplasts ($C_i$-$C_c$, µmol mol⁻¹) and mesophyll conductance to
$CO_2$ ($g_m$, mol $CO_2$ m⁻² s⁻¹); (c) $CO_2$ concentration gradient between ambient air and chloroplasts ($C_a$-$C_c$, µmol mol⁻¹) and total conductance to
$CO_2$ ($g_t$, mol $CO_2$ m⁻² s⁻¹); and (d) $C_a$-$C_c$ and the maximum carboxylase activity of Rubisco ($V_{cmax}$, µmol $CO_2$ m⁻² s⁻¹).








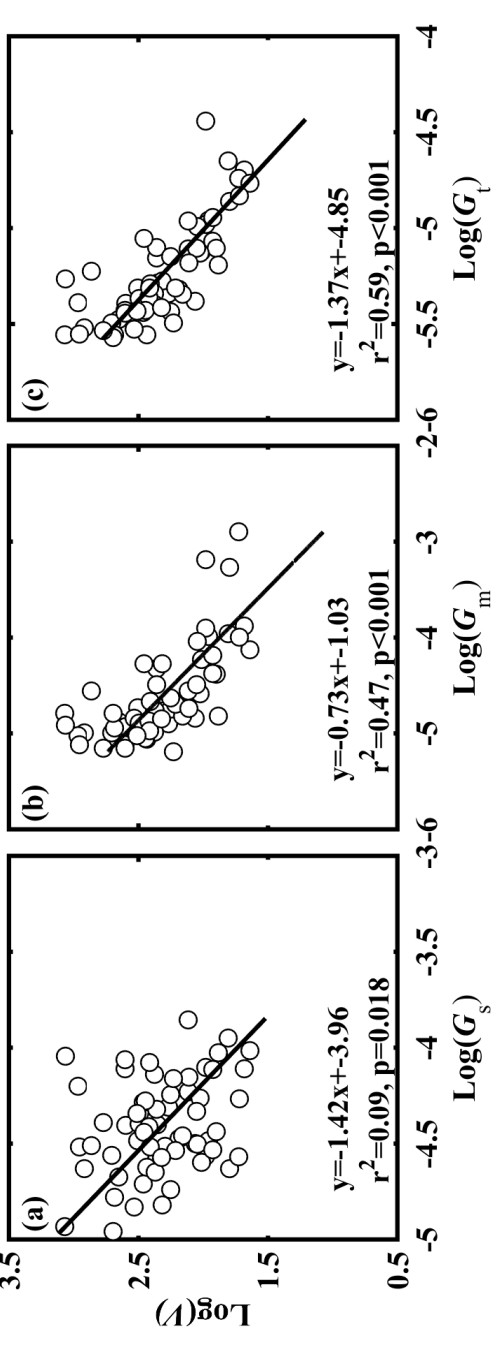


Figure 2. Relationships between (a) $V$ and $G_s$; (b) $V$ and $G_m$; and (c) $V$ and $G_t$. $V$ is the ratio of photosynthetic capacity ($V_{cmax}$) to light-saturated
net photosynthesis ($A$, μmol $CO_2$ m$^{-2}$ s$^{-1}$); $G_s$ is the ratio of stomatal conductance to $CO_2$ ($g_s$, mol $CO_2$ m$^{-2}$ s$^{-1}$) to $A$; $G_m$ is the ratio of mesophyll
conductance to $CO_2$ ($g_m$, mol $CO_2$ m$^{-2}$ s$^{-1}$) to $A$; $G_t$ is the ratio of total conductance to $CO_2$ ($g_t$, mol $CO_2$ m$^{-2}$ s$^{-1}$) to $A$.






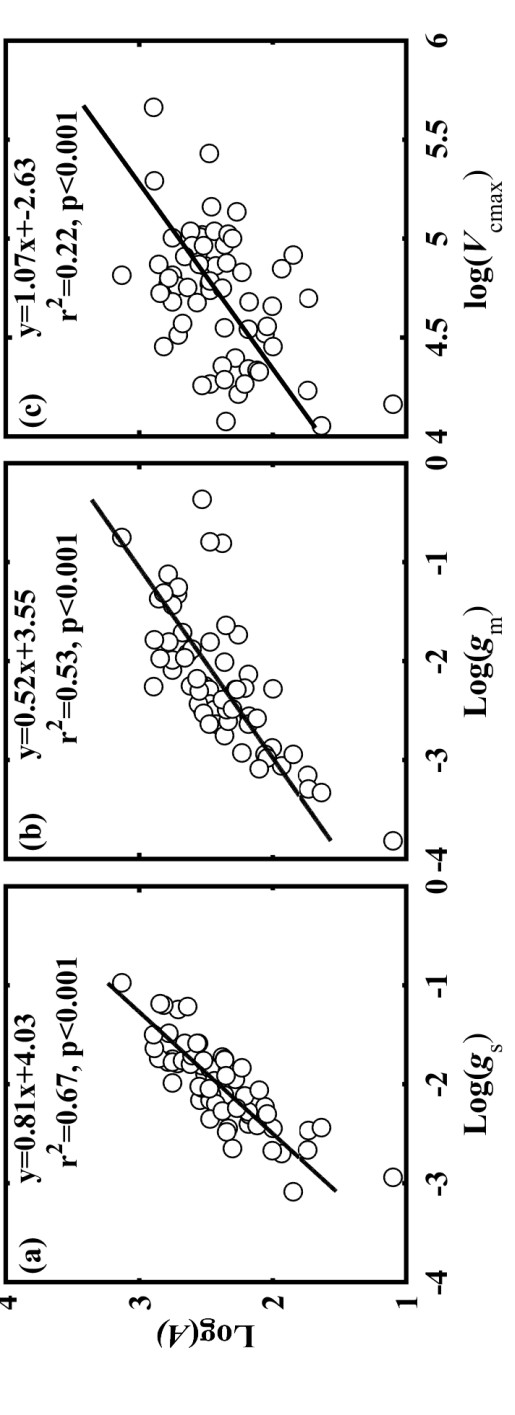

Figure 3. Relationships between light-saturated net photosynthesis ($A$, µmol $CO_2$ m$^{-2}$ s$^{-1}$) and (a) stomatal conductance to $CO_2$ ($g_s$, mol $CO_2$ m$^{-2}$
s$^{-1}$); (b) mesophyll conductance to $CO_2$ ($g_m$, mol $CO_2$ m$^{-2}$ s$^{-1}$); and (c) the maximum carboxylase activity of Rubisco ($V_{cmax}$, µmol $CO_2$ m$^{-2}$ s$^{-1}$).





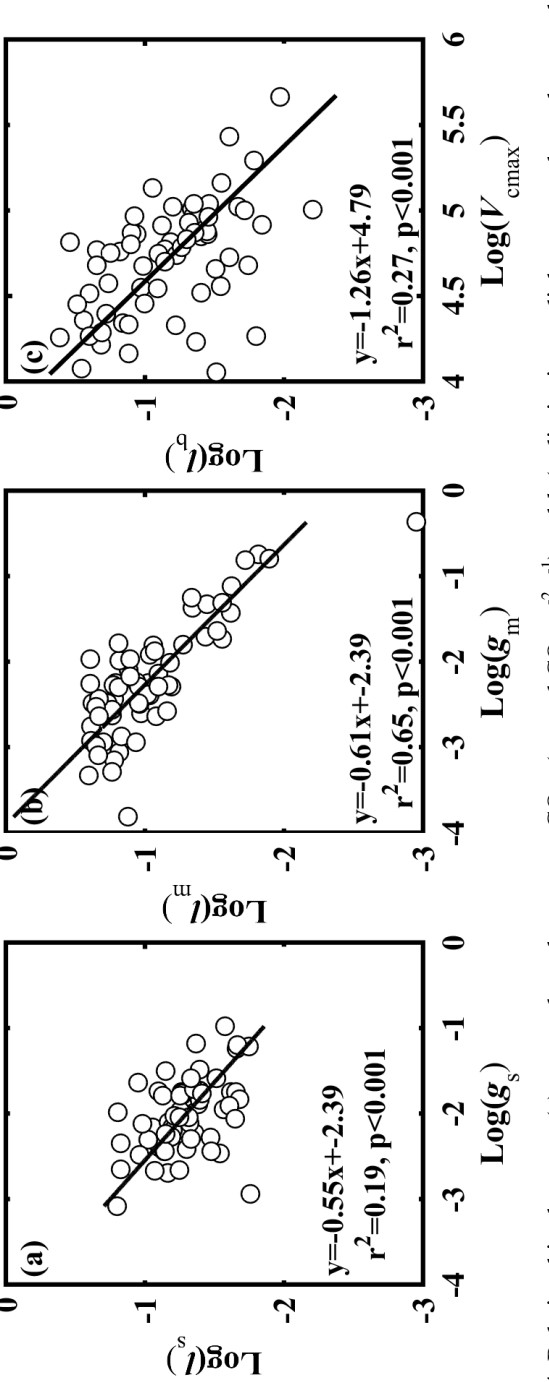


Figure 4. Relationships between (a) stomatal conductance to $CO_2$ ($g_s$, mol $CO_2$ m$^{-2}$ s$^{-1}$) and $l_s$ ($g_s$ limitation on light-saturated net photosynthesis

(A)); (b) mesophyll conductance to $CO_2$ ($g_m$, mol $CO_2$ m$^{-2}$ s$^{-1}$) and $l_m$ ($g_m$ limitation on A); and (c) the maximum carboxylase activity of Rubisco
($V_{cmax}$, μmol $CO_2$ m$^{-2}$ s$^{-1}$) and $l_b$ ($V_{cmax}$ limitation on A).





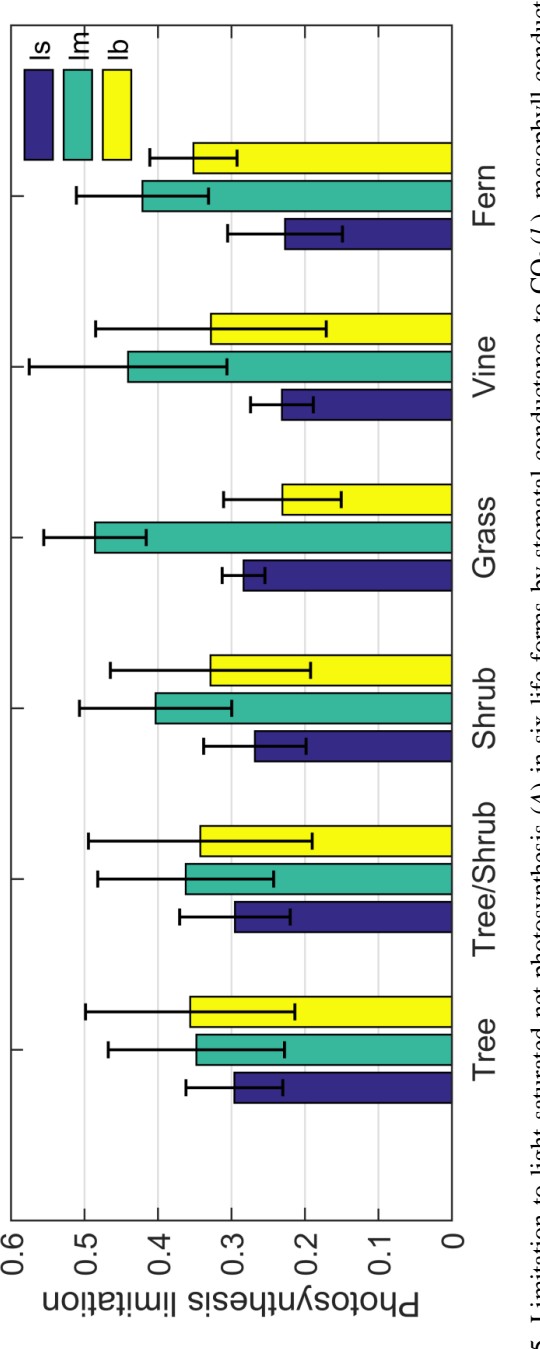

Figure 5. Limitation to light-saturated net photosynthesis ($A$) in six life forms by stomatal conductance to $CO_2$ ($l_s$), mesophyll conductance to $CO_2$ ($l_m$), and the maximum carboxylase activity of Rubisco ($l_b$).






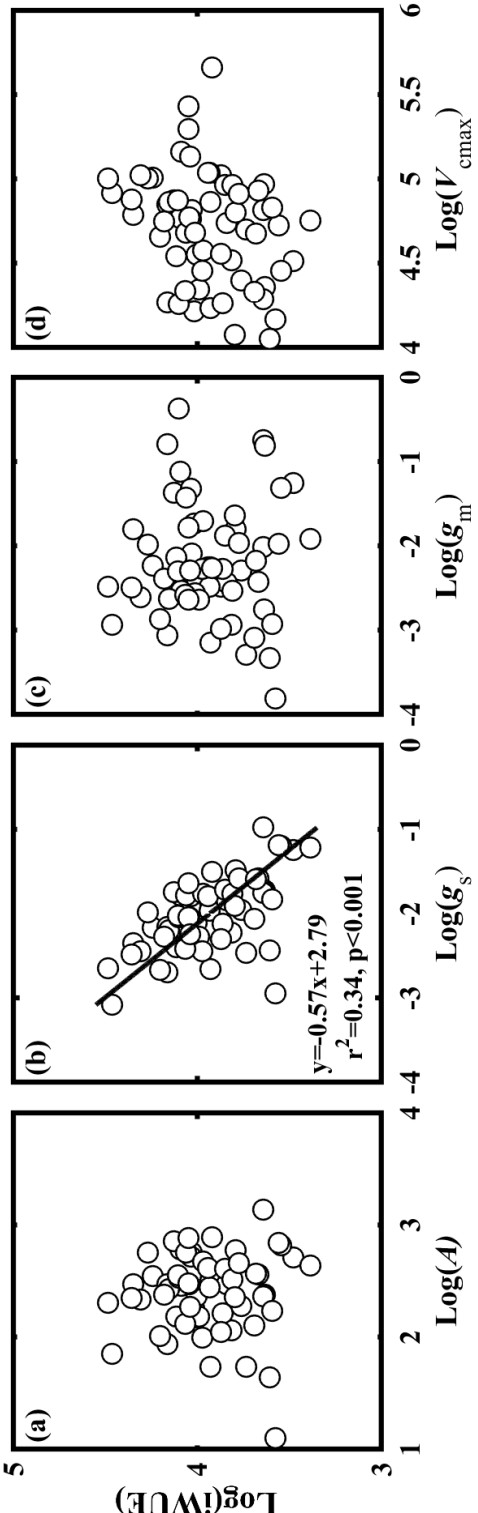

Figure 6. Relationships between the observed intrinsic water use efficiency (iWUE, µmol $CO_2$ mol$^{-1}$ $H_2O$) and (a) light-saturated net

photosynthesis (A, µmol $CO_2$ m$^{-2}$ s$^{-1}$); (b) stomatal conductance to $CO_2$ ($g_s$, mol $CO_2$ m$^{-2}$ s$^{-1}$); (c) mesophyll conductance to $CO_2$ ($g_m$, mol $CO_2$
m$^{-2}$ s$^{-1}$) and (d) the maximum carboxylase activity of Rubisco ($V_{cmax}$, µmol $CO_2$ m$^{-2}$ s$^{-1}$).






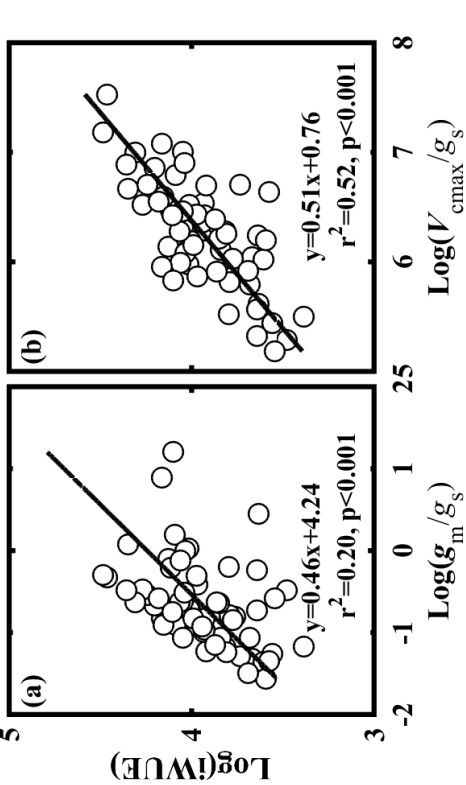


Figure 7. The relationships of the intrinsic water use efficiency (iWUE, $\mu$mol $CO_2$ mol$^{-1}$ $H_2O$) and (a) the ratio of mesophyll conductance to $CO_2$
($g_m$) to ($g_s$) ($g_m/g_s$) and (b) the ratio of the maximum carboxylase activity of Rubisco ($V_{cmax}$) to gs ($V_{cmax}/g_s$).