# Peer review of "The strategies of water-carbon regulation of plants in a subtropical"

_Biogeosciences, 2018_

## Referee Comment (RC1) · Anonymous Referee #1 · 6 Mar 2018

Review for "Trade-offs between water loss and carbon gain in a subtropical primary forest on Karst soils in China"

General comments This is over-all a good article studying gas-exchange and intrinsic water use efficiency relations in a large sample of Karst species. The main results interestingly found that although the area has low-nutrient soil and low water availability, the species had relatively high assimilation rates and low water use efficiency. These were controlled by stomatal conductance, mesophyll conductance and the maximum carboxylase activity of Rubisco and their covariation. The paper is sufficient in detail and has novel insight into an ecosystem that has not been well studied.

Specific comments • I feel the explanation and justification of the chosen methodology for measuring and calculating mesophyll conductance should be in the Materials

and Methods section, not in the discussion. It takes away from your actual results. • Although an "in review" article is cited in the materials and methods, I think this is not an acceptable description of methodology (line 140). This should be written out in detail as I cannot access the information from there. I would like to have more details about leaf sampling and measurements. What were the temperature and humidity chosen for the measurements? How were the leaves collected? ÂăDid you collect leaves or twigs which you then cut under water or did you collect separate leaves which you measured in the field? Did you measure fluorescence? Could you calculate your results with the Harley method as well? It is common nowadays to confirm your results with a second method as all methods have some constraints. • I would also like to see more detail and justification in the statistical analysis section of the materials and methods • In the results, you bring out that gs was better correlated with A, but lm was more limiting. This would be important to discuss in detail in the discussion. This is an extremely important result. • The conclusions are a bit flat, I would like to see the paragraph rephrased so it is a bit more exciting. • Figure 5 needs an explanation about the whiskers: are they SEs or SDs? If they are SEs, I do not find it likely that gm was indeed the most important limiter in vies and ferns, but only grasses.

Technical comments • Line 31: grammatical error, should be "plants'" • Line 38: delete first "and" • Line 38: add "their" between "measured" and "CO2" • Line 38: . . . calculated "the" corresponding. . . • Line 73: replace "indeed" with "however" • Line 84: within "a" leaf. • Line 110: delete "The". Sentences should not be started with an article before an abbreviation. This is bad style. • Lines 125 and 126: this sentence should be in the present if the soil conditions are unlikely to radically change in a short period of time. • Line 130: same comment as the previous, should be in the present if this does not change rapidly. • Line 140: You cannot use "were" if the article you are citing is still in review. This is chronologically incoherent. • Line 148: the citation is doubles, delete one • Line 153: delete "The" • Line 161: no need to redefine abbreviations in each section – once is enough • Line 166: this sentence needs to be rephrased. Stomata are

not a barrier inside the leaf, like this sentence seems to claim. • Line 214: last equation was 8, this should be 9 • Line 253: both implies 2 variables: delete "both of" • Line 256: delete "The" • Line 257: move "respectively" to the end of the sentence • Line 269: delete "The" • Line 271: delete "The" • Line 272: Change to "Grasses" • Line 273: Change to "Accordingly, grasses" • Line 276: delete "The" • Line 284: delete "The" • Line 295: Recent work has compared Harley, Ethier and the anatomical models finding good correlations, so I would not write largely unknown, rather "to some extent" • Line 353: this sentence should be rephrased, leads to the impression that you also did ultrastructural sampling • Lines 368-374: chloroplasts do not have cell walls, the sentences need to be rephrased • Line 402: "highly efficient" • Line 411: delete the first "in this study" • Line 415: "lose" not "loss" • Lines 416-417 "The results . . .": unnecessary sentence, delete • Line 422: full stop missing from the end • Line 424: delete "The" • Lines 424-425 stating with "In theory": should be in the present • Line 433: This sentence should be in the present • Line 448: . . .inefficiency in "the" trade-off • Line 452: "low nutrient" • Line 461: iWUE is not in italic in any other place • Line 462: . . .forms in "the" field • Line 463: . . . used "a" diverse • Line 464: . . . maintain "a" relatively • Line 465: . . . used "the" • Line 483: "References"

Please also note the supplement to this comment:
https://www.biogeosciences-discuss.net/bg-2018-44/bg-2018-44-RC1-supplement.pdf

---

## Referee Comment (RC2) · Anonymous Referee #2 · 5 Apr 2018

The strategy of water-carbon regulation of plants in water stressed region, e.g. karst area, is an interesting topic. This study measured 189 A-Ci curves for 63 species and try to answer the limitations of CO2 diffusion (gs and gm) and Vcmax on A and iWUE in different life forms. Considering that the authors collected many valuable data of different life form plants, it's worth publishing. The topic is not very new but still interesting if can the data can be well organized and presented. However, the results are not well analyzed and presented. the discussion didn't focus on the title and there are many other issues that authors should substantially improve on the current manuscript. General comments:

1. The author use "Trade-offs between water loss and carbon gain" in the title, however, the whole-text actually talk about the limitation of different components on A and iWUE.

[Figure]

It's better to rephrase the title.

2. In the method section: The species covered wide range of functional groups, including 6 life forms. What the criteria of the species selection? Because the leaf habit (evergreen or deciduous), the shade or light-demanding behaviors also will affect the strategy of plant carbon-water regulation. For example, does fern grow in the canopy or understory, how you can put them together when analyze the data?

More important, the main objective of this paper was to determine and distinguish the limitations of $CO_2$ diffusion and $V_{cmax}$ on A and iWUE in different life forms Karst forest, however, you combine all species together for most analysis, actually we do not know what's the difference between different life forms in Figs 1-4, 6,7. I Believe most land plant will behave in similar way to adapt to the environmental factor no matter where they grow, the interesting things is to what extent by different plants. For example, Based on Fig 5, we could not see any difference among the groups. So, I suggest the author should separate into 6 groups to see the differences of regression lines among groups for all the figures, and compare the difference among the life forms using proper statistical method.

3. lines 139-140, because the A-Ci curve is the key data of this paper, author should describe in detail how this measurement was done rather than just cite other submitted papers. For example, you should introduce the he,ight of your targeted individuals? how you can measure the sun-exposed leaf for canopy trees and climbing plants….? did you measure in situ or cut down, if the latter, for A-Ci curve you normally need ca. 30 min, how you can avoid the effects of cutting on stomatal conductance because some species are very sensitive, do you have some information on the gs sensitivity for those speciesïij§…..

Specific comments:

1. Line 267-269: There is no statistic tests of the differences of the results in figure 5, so it is not proper to give the statements in line 309-310. Figure 5 can't give any information that is about LMA. Please use data to demonstrate the relationship between LMA and other parameters instead of qualitative description.

2. Line 372: Species with low LMA may have thick cell walls in mesophyll and chloroplast.

3. Line 381-382: In your results, gs and gm are positively correlated, why did you conclude gm is a compensate for reductions in gs? Did you observe an increasing of gm when gs decreased.

4. Line 384-389: I don't think you have enough evidences to state "there was a trend of increasing lm with increasing leaf N:P", unless you add this part of research in your draft.

5. Awful sentences, Lines 39-35, should split into short sentences

---

## Author Comment (AC1) · 25 Apr 2018

Dear Reviewer,

We would like to thank you for the thoughtful and valuable comments and suggestions on our manuscript entitled "Trade-offs between water loss and carbon gain in a subtropical primary forest on Karst soils in China" (bg-2018-44). We have carefully revised our manuscript to take account of your comments and suggestions. Please find below our responses (upright Roman) to comments (original queries in Italic). Meanwhile, we have rephrased our manuscript title as "The strategies of water-carbon regulation of plants in a subtropical primary forest on Karst soils in China". The line numbers mentioned here refer to our original manuscript. The changed figures and tables are

presented in the Appendix (listed at the end of the "Response to reviewer").

Specific comments (1) I feel the explanation and justification of the chosen methodology for measuring and calculating mesophyll conductance should be in the Materials and Methods section, not in the discussion. It takes away from your actual results.

Response: Thank you for your suggestion. This section have been moved to Section "Materials and Methods" according to your suggestion.

(2) Although an "in review" article is cited in the materials and methods, I think this is not an acceptable description of methodology (line 140). This should be written out in detail as I cannot access the information from there. I would like to have more details about leaf sampling and measurements. What were the temperature and humidity chosen for the measurements? How were the leaves collected? Did you collect leaves or twigs which you then cut under water or did you collect separate leaves which you measured in the field? Did you measure fluorescence? Could you calculate your results with the Harley method as well? It is common nowadays to confirm your results with a second method as all methods have some constraints.

Response: Thank you for your suggestions. In response, we have revised the Section "Materials and Methods" in two aspects. Firstly, we added more details about leaf sampling and measurements in Section "Materials and Methods". Such as, we have added the method of how were the leaves collected and prepared before $CO_2$ response curves measurements "Branches exposed to the sun were excised from the upper part of the crown (Trees, Tree/Shrubs, Shrubs and Vines) or aboveground portion (Grasses, Ferns), and immediately re-cut under water to maintain xylem water continuity. Back into the laboratory, branches and aboveground portions were kept at 25oC for 30 min. Fully-expanded and mature leaves were induced for 30 minutes at a saturating light density (1500 $\mu$mol m-2 s-1). $CO_2$ response curves measurements were performed when A and gs was stable. Three leaves per species were collected and measured. A total of 189 leaves were collected from adult individuals of 63 species."

We have described the method and conditions of CO2 response curves measurements in more detail as: "The CO2 response curves were measured with 11 CO2 concentration gradients in chamber following the procedural guidelines described by Longand Bernacchi (2003). The photosynthetic photon flux density was 1500 $\mu$mol m–2 s–1. The leaf temperature was 25°C, controlled by the block temperature. The humidity in the leaf chamber was maintained at ambient condition. Leaf area, thickness (LT) and dry mass were measured after the CO2 response measurements. Leaf mass per area (LMA) was calculated by dividing the corresponding dry mass by leaf area. And leaf density (LD) was calculated by dividing the corresponding LMA by LT. More details were described in Wang et al. (2018)."

Secondly, we clarified that gm was estimated by the 'curve-fitting' method in this study. As the fluorescence was not measured in this study, the Harley method cannot be used to calculate gm. Details about why we choose the 'curve-fitting' method to calculate gm, and the data valid confirmation have been added "Three methods are most commonly used for gm estimation. Those methods have been reviewed by Warren (2006) and Pons et al. (2009). Briefly, gm can be calculated by the stable isotope method (Evans, 1983; Sharkey et al., 1991; Loreto et al., 1992), J method (Bongi and Loreto, 1989; Dimarco et al., 1990; Harley et al., 1992; Epron et al., 1995; Laisk et al., 2005), and 'curve-fitting' method (Ethier and Livingston, 2004; Sharkey et al., 2007). All of these methods are based on gas exchange measurements (Pons et al., 2009), and some common assumptions (Warren, 2006). Thus, the accuracy of each method is to some extent unknown (Warren, 2006). gm was estimated by the 'curve-fitting' method in this study. Although the 'curve-fitting' method is less precise than the stable isotope method, the 'curve-fitting' method is much more readily available and has been used for several decades (Warren, 2006; Sharkey, 2012). Accurate measurements of A and Ci is a prerequisite for estimating gm using the 'curve-fitting' method (Pons et al., 2009). Warren (2006) pointed out that highly-accurate measurements need small leaf area and low flow rates. We confirmed that the calculated Cc and the initial slope of A-Cc curves were positive, suggesting that the measured gm was reliable (Warren,

2006). "

(3) I would also like to see more detail and justification in the statistical analysis section of the materials and methods

Response: Thank you for your comment. In response, we have moved the Section "2.4 Quantitative analysis of limitations on A" to Section "2.5 Statistical analysis" as the first section. Meanwhile, we have added more data analysis details in Section "2.5 Statistical analysis" as the second section. Such as, we have added the data analysis method "Data were analyzed either as a whole group (six life forms combined) or by individual life forms.". We have added the bivariate linear regressions method "The bivariate linear regressions of leaf gas exchange parameters were performed using the standardized major axis (SMA) regression fits, and all of the data were made on loge-transformed data (Table S2).". We have added what method was used to compare the difference of linear regressions "To test for the differences among life forms, SMA regression fits were used to compare the slope of regression lines which significant relationships had already been obtained. Note that Grass, Vine and Fern were not considered due to the small sample size. A similar trend was obtained, and no significant difference was found between life forms although significant relationships were not obtained for some bivariate linear regressions. Accordingly, six life forms were grouped together to analyze the strategy of water-carbon regulation of plants in the whole text. ". We have added what method was used to compare the difference of the relative limitations of gs, gm and Vcmax to A. " The difference of the contribution of gs, gm and Vcmax to A among life forms or as a whole group were performed using one-way ANOVA and Duncan multiple comparison. The probability of significance was defined at p< 0.05. "

(4) In the results, you bring out that gs was better correlated with A, but lm was more limiting. This would be important to discuss in detail in the discussion. This is an extremely important result.

Response: Thank you for your comment and suggestion. In response, we have rean-alyzed our data, and revised Section "4.1 Co-variation in gs, gm and Vcmax in reg-ulating A". Firstly, we analyzed the relationships between $CO_2$ diffusion conductance (gs and gm) and Vcmax, compared the relative limitations of gs, gm and Vcmax to A, and analyzed the relationships between the limitation factors and the corresponding relative limitations. Consequently, we have revised the paragraph in Section "4.1 Co-variation in gs, gm and Vcmax in regulating A" "The A was constrained by gs, gm, and Vcmax acting together, however, variability in the relative contribution of these three factors depended on species and habitats (Tosens et al., 2016; Galmes et al., 2017; Peguero-Pina et al., 2017a; Veromann-Jurgenson et al., 2017). .... In addition, 20 of the 63 species were mainly limited by Vcmax (lb>0.4, with the largest value of 0.68). (lines 340-351)" to "A was constrained by gs, gm, and Vcmax acting together, how-ever, variability in the relative contribution of these three factors depended on species and habitats (Tosens et al., 2016; Galmes et al., 2017; Peguero-Pina et al., 2017a; Veromann-Jurgenson et al., 2017). A was significantly correlated with gs, gm, and Vc-max (Fig.3a-c). gs was positively related to gm (Fig.S1c), while no relationship was found between the $CO_2$ diffusion conductance (gs and gm) and Vcmax (Fig. S2). The relative limitations of gs, gm, and Vcmax were separated by a quantitative limitation model (Jones, 1985; Grassi & Magnani, 2005). The results showed that ls, lm and lb of 63 species varied in a large range (Fig. S3), indicating plants have a diverse strategies to co-ordinate the $CO_2$ diffusion (gs and gm) and Vcmax to maintain relative high A. The order of factors limitations to A was lm> lb >ls (P<0.05) (Fig.S3). Furthermore, we tested the relationship between the relative limitations and the corresponding lim-itation factors. The results showed that ls, lm, and lb were negatively associated with gs, gm, and Vcmax, respectively (Fig. 4). And the relationship was stronger for gm- lm (r2=0.65) than Vcmax- lb (r2=0.27) and gs- ls (r2=0.19). "

Secondly, we have discussed two possible reasons of the results in Section "4.1 Co-variation in gs, gm and Vcmax in regulating A". "gs was better correlated with A, while the results showed that A was more limited by gm. That could be explained by two possible reasons. Firstly, compare to the linear relationship between A and gs, a nonlinear trend has been found between A and gm when gm>0.4 (Fig. 3a, b). Secondly, leaf structure plays an important role in regulating gm and Vcmax, consequently, in determining A (Veromann-Jurgenson et al., 2017). Negative relationships between A/LMA and LT (r2=0.16, p=0.002), and A/LMA and LT (r2=0.3, p<0.001) have been observed (Fig. S4c,d), while A was not correlated to LT and LD (Fig. S4a,b).

The importance of gm in constraining A was variable, and depended on leaf structural traits, only LMA, LT, and LD were analyzed in this study. Large variability in gm has been shown both between and within species with different life forms and habits (Gago et al., 2014; Flexas et al., 2016). Variability in gm in this study is similar to that in global datasets (Gago et al., 2014; Flexas et al., 2016). There was no significantly difference among life forms (P>0.05). Previous studies have confirmed that LMA (Tomas et al., 2013), thickness of leaf cell wall (Peguero-Pina et al., 2017b), liquid phase of mesophyll (Veromann-Jurgenson et al., 2017), cell wall thickness of mesophyll (Terashima et al., 2011;Tosens et al., 2016), and surface area of mesophyll and chloroplast exposed to intercellular space (Veromann-Jurgenson et al., 2017) were the main limitations for gm. The wide variability of gm between different species and life forms in the same ecosystem seems to be related to the diversity of leaf anatomical traits.

No significant difference of LMA, LT, and LD was found among life forms (P<0.05). The negative correlation of gm (Terashima et al., 2005) or gm/LMA (Niinemets et al., 2009; Veromann-Jurgenson et al., 2017) with LMA have been reported. In this study, there was a significant relationship between gm/LMA with LMA (P<0.01), however, no relationship was found between gm with LMA. gm/LMA was significantly negative related to LD (p<0.01) (Fig. S5c), and weak negative related to LT (p=0.06) (Fig. S5d), demonstrating that the negative role of cell wall thickness on gm (Terashima et al., 2006; Niinemets et al., 2009). The strong investment in supportive structures was the main reason for the limitation of gm on A (Veromann-Jurgenson et al., 2017). However, it is still unknown how leaf anatomical traits affect gm and A, and this should be further

explored.

gs is responsible for CO2 exchange between atmosphere and leaf, and regulate the CO2 fixation (A) and water loss (Lawsonand Blatt, 2014). The variability of gs was controlled by stomatal anatomy, i.e. stomata density and size, and mesophyll demands for CO2 (Lawsonand Blatt, 2014). However, the stomatal anatomy was not analyzed in this study. We only focused on how the relationship between gs and gm regulate A. Positive relationship between gs and gm has been observed (Flexas et al., 2013). For example. the restricted CO2 diffusion from the ambient air to chloroplast is the main reason for a decreased A under water stress conditions due to both the stomatal and mesophyll limitations (Olsovska et al., 2016). gs was significantly positive related to gm for 63 species (P<0.001, Fig. S1) in this study, and no difference of the slopes of regression lines between gs and gm was found among life forms, demonstrating that A was regulated by the co-variation of gs and gm. However, the variability of gm and lm was larger than gs and ls, respectively (Fig.1 and Fig.S3).

The wide variation range of lb (0.11-0.68) highlighted the importance role of Vcmax in regulating A. Vcmax was used to represent the CO2 demand in photosynthetic process in this study. The relative contribution of Vcmax to A not only depends on Ca-Cc, but also on leaf nutrient levels. Positive relationship was found between Ca-Cc and Vcmax (Fig. 1d). And the Vcmax/LMA was co-regulated by leaf N, P and Mg content (Jing et al. 2018). In addition, Vcmax/LMA was negatively related to LT (p<0.05) (Fig. S6c) and LD (p<0.05) (Fig. S6d), while Vcmax was not correlated to LT and LD (Fig. S6a,b), demonstrating that leaf structure plays an important role in regulating Vcmax."

(5) The conclusions are a bit flat, I would like to see the paragraph rephrased so it is a bit more exciting.

Response: Thank you for your comment. The Section "Conclusions" has been rephrased as: "This study provides information of limitations of A and iWUE by gs, gm, and Vcmax in 63 species across 6 life forms in the field. The results showed that

plants growing in Karst CZs used a diverse strategies of carbon-water regulation, but no difference was found among life forms. The co-variation of CO2 supply (gs and gm) and demand (Vcmax) regulated A, indicating that species maintain relative high A through co-varing their leaf anatomical structure and Vcmax. iWUE was relatively low, but ranged widely, indicating that plants used the 'profligate/opportunistic' water use strategy to maintain the survival, growth, and structure of the community. iWUE was regulated by gs, Vcmax, gm/gs and Vcmax/gs, indicating that species with high gm/gs or Vcmax/gs will have to be much more competitive to response to the ongoing rapid warming and drought in the Karst CZs."

(6) Figure 5 needs an explanation about the whiskers: are they SEs or SDs? If they are SEs, I do not find it likely that gm was indeed the most important limiter in vies and ferns, but only grasses.

Response: Thank you for your suggestion and comment. We clarified that whickers in Figure 5 was standard deviation. The Figure 5 legend rephrased as: "Figure 5. Limitation to light-saturated net photosynthesis (A) in six life forms by stomatal conductance to CO2 (ls), mesophyll conductance to CO2 (lm), and the maximum carboxylase activity of Rubisco (lb). Error bars denominate standard deviation.".

Technical comments

(7) Line 31: grammatical error, should be "plants"'

Response: Corrected. Thank you.

(8) Line 38: delete first "and"

Response: Deleted. Thank you.

(9) Line 38: add "their" between "measured" and "CO2"

Response: Change has been made. Thank you.

(10)Line 38: . . . calculated "the" corresponding. . .

Response: Change has been made. Thank you.

(11) Line 73: replace "indeed" with "however"

Response: This change has been made.

(12) Line 84: within "a" leaf.

Response: Change has been made. Thank you.

(13) Line 110: delete "The". Sentences should not be started with an article before an abbreviation. This is bad style.

Response: Deleted. Thank you.

(14) Lines 125 and 126: this sentence should be in the present if the soil conditions are unlikely to radically change in a short period of time.

Response: Change has been made.

(15) Line 130: same comment as the previous, should be in the present if this does not change rapidly.

Response: Change has been made.

(16) Line 140: You cannot use "were" if the article you are citing is still in review. This is chronologically incoherent.

Response: Thank you for your suggestion and comment. The cited article has been accepted by Scientific Reports". And this sentence has been rephrased as "More details were described in Wang et al. (2018)."

(17) Line 148: the citation is doubles, delete one

Response: Deleted. Thank you. (See page 7 line 174)

(18) Line 153: delete "The"

Response: Deleted. Thank you.

(19) Line 161: no need to redefine abbreviations in each section – once is enough

Response: Change has been made.

(20) Line 166: this sentence needs to be rephrased. Stomata are not a barrier inside the leaf, like this sentence seems to claim.

Response: Thank you for your suggestion and comment. Rephrased as: "Mesophyll is the barrier for CO2 inside the leaf. "

(21) Line 214: last equation was 8, this should be 9

Response: This changed have been made. Thank you.

(22)Line 253: both implies 2 variables: delete "both of"

Response: Deleted.

(23) Line 256: delete "The"

Response: Deleted.

(24) Line 257: move "respectively" to the end of the sentence

Response: Change has been made. Thank you.

(25) Line 269: delete "The"

Response: Deleted.

(26) Line 271: delete "The"

Response: Deleted.

(27) Line 272: Change to "Grasses"

Response: Change has been made. Thank you.

(28) Line 273: Change to "Accordingly, grasses"

Response: Change has been made.

(29) Line 276: delete "The"

Response: Deleted.

(30) Line 284: delete "The"

Response: Deleted.

(31) Line 295: Recent work has compared Harley, Ethier and the anatomical models finding good correlations, so I would not write largely unknown, rather "to some extent"

Response: Rephrased as: "Thus, the accuracy of each method is to some extent unknown (Warren, 2006). "

(32) Line 353: this sentence should be rephrased, leads to the impression that you also did ultrastructural sampling

Response: Thank you for your suggestion and comment. Rephrased as: "The importance of gm in constraining A was variable, and depended on leaf structural traits, only LMA, LT, and LD were analyzed in this study."

(33)Lines 368-374: chloroplasts do not have cell walls, the sentences need to berephrased

Response: Thank you for your suggestion and comment. This mistake has been corrected.

(34) Line 402: "highly efficient"

Response: This change has been made.

(35)Line 411: delete the first "in this study"

Response: Deleted.

(36)Line 415: "lose" not "loss"

Response: Corrected. Thank you.

(37) Lines 416-417 "The results . . .": unnecessary sentence, delete

Response: Deleted.

(38)Line 422: full stop missing from the end

Response: Added. Thank you.

(39) Line 424: delete "The"

Response: This change has been made.

(40) Lines 424-425 stating with "In theory": should be in the present

Response: This change has been made.

(41) Line 433: This sentence should be in the present

This change has been made.

(42) Line 448: . . .inefficiency in "the" trade-off

Response: This change has been made.

(43) Line 452: "low nutrient"

Response: This change has been made.

(44) Line 461: iWUE is not in italic in any other place

Response: This change has been made.

(45)Line 462: . . .forms in "the" field

Response: This change has been made.

(46) Line 463: . . . used "a" diverse

Response: Change has been made, thank you.

(47) Line 464: . . . maintain "a" relatively

Response: This change has been made.

(48) Line 465: . . . used "the"

Response: Thank you for your suggestion and comment. Chang has been made.

(49)Line 483: "References"

Response: Change has been made.

Please also note the supplement to this comment:
https://www.biogeosciences-discuss.net/bg-2018-44/bg-2018-44-AC1-supplement.pdf

**Supplement:**

Response to reviews of manuscript "Trade-offs between water loss and carbon gain in a subtropical primary forest on Karst soils in China" bg-2018-44

**Response to reviewer**

Dear Reviewer,

We would like to thank you for the thoughtful and valuable comments and suggestions on our manuscript entitled "Trade-offs between water loss and carbon gain in a subtropical primary forest on Karst soils in China" (bg-2018-44). We have carefully revised our manuscript to take account of your comments and suggestions. Please find below our responses (upright Roman) to comments (original queries in Italic). Meanwhile, we have rephrased our manuscript title as "The strategies of water-carbon regulation of plants in a subtropical primary forest on Karst soils in China". The line numbers mentioned here refer to our original manuscript. The changed figures and tables are presented in the Appendix (listed at the end of the "Response to reviewer").

Specific comments

*(1) I feel the explanation and justification of the chosen methodology for measuring and calculating mesophyll conductance should be in the Materials and Methods section, not in the discussion. It takes away from your actual results.*

Response: Thank you for your suggestion. This section have been moved to Section "Materials and Methods" according to your suggestion.

*(2) Although an "in review" article is cited in the materials and methods, I think this is not an acceptable description of methodology (line 140). This should be written out in detail as I cannot access the information from there. I would like to have more details about leaf sampling and measurements. What were the temperature and humidity chosen for the measurements? How were the leaves collected? Did you collect leaves or twigs which you then cut under water or did you collect separate*

*leaves which you measured in the field? Did you measure fluorescence? Could you calculate your results with the Harley method as well? It is common nowadays to confirm your results with a second method as all methods have some constraints.*

Response: Thank you for your suggestions. In response, we have revised the Section "Materials and Methods" in two aspects. Firstly, we added more details about leaf sampling and measurements in Section "Materials and Methods". Such as, we have added the method of how were the leaves collected and prepared before $CO_2$ response curves measurements "Branches exposed to the sun were excised from the upper part of the crown (Trees, Tree/Shrubs, Shrubs and Vines) or aboveground portion (Grasses, Ferns), and immediately re-cut under water to maintain xylem water continuity. Back into the laboratory, branches and aboveground portions were kept at 25$^o$C for 30 min. Fully-expanded and mature leaves were induced for 30 minutes at a saturating light density (1500 µmol m$^{-2}$ s$^{-1}$). $CO_2$ response curves measurements were performed when $A$ and $g_s$ was stable. Three leaves per species were collected and measured. A total of 189 leaves were collected from adult individuals of 63 species."

We have described the method and conditions of $CO_2$ response curves measurements in more detail as: "The $CO_2$ response curves were measured with 11 $CO_2$ concentration gradients in chamber following the procedural guidelines described by Longand Bernacchi (2003). The photosynthetic photon flux density was 1500 µmol m$^{-2}$ s$^{-1}$. The leaf temperature was 25 ℃, controlled by the block temperature. The humidity in the leaf chamber was maintained at ambient condition. Leaf area, thickness (LT) and dry mass were measured after the $CO_2$ response measurements. Leaf mass per area (LMA) was calculated by dividing the corresponding dry mass by leaf area. And leaf density (LD) was calculated by dividing the corresponding LMA by LT. More details were described in Wang et al. (2018)."

Secondly, we clarified that $g_m$ was estimated by the 'curve-fitting' method in this study. As the fluorescence was not measured in this study, the Harley method cannot be used to calculate $g_m$. Details about why we choose the 'curve-fitting' method to

calculate $g_m$, and the data valid confirmation have been added "Three methods are most commonly used for $g_m$ estimation. Those methods have been reviewed by Warren (2006) and Pons et al. (2009). Briefly, $g_m$ can be calculated by the stable isotope method (Evans, 1983; Sharkey et al., 1991; Loreto et al., 1992), $J$ method (Bongi and Loreto, 1989; Dimarco et al., 1990; Harley et al., 1992; Epron et al., 1995; Laisk et al., 2005), and 'curve-fitting' method (Ethier and Livingston, 2004; Sharkey et al., 2007). All of these methods are based on gas exchange measurements (Pons et al., 2009), and some common assumptions (Warren, 2006). Thus, the accuracy of each method is to some extent unknown (Warren, 2006).

$g_m$ was estimated by the 'curve-fitting' method in this study. Although the 'curve-fitting' method is less precise than the stable isotope method, the 'curve-fitting' method is much more readily available and has been used for several decades (Warren, 2006; Sharkey, 2012). Accurate measurements of $A$ and $C_i$ is a prerequisite for estimating $g_m$ using the 'curve-fitting' method (Pons et al., 2009). Warren (2006) pointed out that highly-accurate measurements need small leaf area and low flow rates. We confirmed that the calculated $C_c$ and the initial slope of $A$-$C_c$ curves were positive, suggesting that the measured $g_m$ was reliable (Warren, 2006). "

*(3) I would also like to see more detail and justification in the statistical analysis section of the materials and methods*

Response: Thank you for your comment. In response, we have moved the Section "2.4 Quantitative analysis of limitations on $A$" to Section "2.5 Statistical analysis" as the first section. Meanwhile, we have added more data analysis details in Section "2.5 Statistical analysis" as the second section. Such as, we have added the data analysis method "Data were analyzed either as a whole group (six life forms combined) or by individual life forms.". We have added the bivariate linear regressions method "The bivariate linear regressions of leaf gas exchange parameters were performed using the standardized major axis (SMA) regression fits, and all of the data were made on $\log_e$-transformed data (Table S2).". We have added what method was used to compare

the difference of linear regressions "To test for the differences among life forms, SMA regression fits were used to compare the slope of regression lines which significant relationships had already been obtained. Note that Grass, Vine and Fern were not considered due to the small sample size. A similar trend was obtained, and no significant difference was found between life forms although significant relationships were not obtained for some bivariate linear regressions. Accordingly, six life forms were grouped together to analyze the strategy of water-carbon regulation of plants in the whole text. ". We have added what method was used to compare the difference of the relative limitations of $g_s$, $g_m$ and $V_{cmax}$ to $A$. " The difference of the contribution of $g_s$, $g_m$ and $V_{cmax}$ to $A$ among life forms or as a whole group were performed using one-way ANOVA and Duncan multiple comparison. The probability of significance was defined at $p< 0.05$. "

*(4) In the results, you bring out that gs was better correlated with A, but lm was more limiting. This would be important to discuss in detail in the discussion. This is an extremely important result.*

Response: Thank you for your comment and suggestion. In response, we have reanalyzed our data, and revised Section "4.1 Co-variation in $g_s$, $g_m$ and $V_{cmax}$ in regulating $A$". Firstly, we analyzed the relationships between $CO_2$ diffusion conductance ($g_s$ and $g_m$) and $V_{cmax}$, compared the relative limitations of $g_s$, $g_m$ and $V_{cmax}$ to $A$, and analyzed the relationships between the limitation factors and the corresponding relative limitations. Consequently, we have revised the paragraph in Section "4.1 
[revised manuscript text omitted]
 conclusions are a bit flat, I would like to see the paragraph rephrased so it is a bit more exciting.*

Response: Thank you for your comment. The Section "Conclusions" has been rephrased as: "This study provides information of limitations of $A$ and iWUE by $g_s$, $g_m$, and $V_{cmax}$ in 63 species across 6 life forms in the field. The results showed that plants growing in Karst CZs used a diverse strategies of carbon-water regulation, but no difference was found among life forms. The co-variation of $CO_2$ supply ($g_s$ and $g_m$) and demand ($V_{cmax}$) regulated $A$, indicating that species maintain relative high $A$ through co-varing their leaf anatomical structure and $V_{cmax}$. iWUE was relatively low, but ranged widely, indicating that plants used the 'profligate/opportunistic' water use strategy to maintain the survival, growth, and structure of the community. iWUE was regulated by $g_s$, $V_{cmax}$, $g_m$/$g_s$ and $V_{cmax}$/$g_s$, indicating that species with high $g_m$/$g_s$ or $V_{cmax}$/$g_s$ will have to be much more competitive to response to the ongoing rapid warming and drought in the Karst CZs."

*(6) Figure 5 needs an explanation about the whiskers: are they SEs or SDs? If they are SEs, I do not find it likely that gm was indeed the most important limiter in vies and ferns, but only grasses.*

Response: Thank you for your suggestion and comment. We clarified that whickers in Figure 5 was standard deviation. The Figure 5 legend rephrased as: "Figure 5. Limitation to light-saturated net photosynthesis ($A$) in six life forms by stomatal conductance to $CO_2$ ($l_s$), mesophyll conductance to $CO_2$ ($l_m$), and the maximum carboxylase activity of Rubisco ($l_b$). Error bars denominate standard deviation.".

Technical comments

*(7) Line 31: grammatical error, should be "plants'"*
Response: Corrected. Thank you.

*(8) Line 38: delete first "and"*
Response: Deleted. Thank you.

*(9) Line 38: add "their" between "measured" and "CO2"*
Response: Change has been made. Thank you.

*(10) Line 38: ... calculated "the" corresponding...*
Response: Change has been made. Thank you.

*(11) Line 73: replace "indeed" with "however"*
Response: This change has been made.

*(12) Line 84: within "a" leaf.*
Response: Change has been made. Thank you.

*(13) Line 110: delete "The". Sentences should not be started with an article before an abbreviation. This is bad style.*
Response: Deleted. Thank you.

*(14) Lines 125 and 126: this sentence should be in the present if the soil conditions are unlikely to radically change in a short period of time.*
Response: Change has been made.

*(15) Line 130: same comment as the previous, should be in the present if this does not change rapidly.*

Response: Change has been made.

*(16) Line 140: You cannot use "were" if the article you are citing is still in review. This is chronologically incoherent.*

Response: Thank you for your suggestion and comment. The cited article has been accepted by "Scientific Reports". And this sentence has been rephrased as "More details were described in Wang et al. (2018)."

*(17) Line 148: the citation is doubles, delete one*

Response: Deleted. Thank you. (See page 7 line 174)

*(18) Line 153: delete "The"*

Response: Deleted. Thank you.

*(19) Line 161: no need to redefine abbreviations in each section – once is enough*

Response: Change has been made.

*(20) Line 166: this sentence needs to be rephrased. Stomata are not a barrier inside the leaf, like this sentence seems to claim.*

Response: Thank you for your suggestion and comment. Rephrased as: "Mesophyll is the barrier for $CO_2$ inside the leaf. "

*(21) Line 214: last equation was 8, this should be 9*

Response: This changed have been made. Thank you.

*(22)Line 253: both implies 2 variables: delete "both of"*

Response: Deleted.

*(23) Line 256: delete "The"*

Response: Deleted.

*(24) Line 257: move "respectively" to the end of the sentence*

Response: Change has been made. Thank you.

*(25) Line 269: delete "The"*

Response: Deleted.

*(26) Line 271: delete "The"*

Response: Deleted.

*(27) Line 272: Change to "Grasses"*

Response: Change has been made. Thank you.

*(28) Line 273: Change to "Accordingly, grasses"*

Response: Change has been made.

*(29) Line 276: delete "The"*

Response: Deleted.

*(30) Line 284: delete "The"*

Response: Deleted.

*(31) Line 295: Recent work has compared Harley, Ethier and the anatomical models finding good correlations, so I would not write largely unknown, rather "to some extent"*

Response: Rephrased as: "Thus, the accuracy of each method is to some extent unknown (Warren, 2006). "

*(32) Line 353: this sentence should be rephrased, leads to the impression that you also did ultrastructural sampling*

Response: Thank you for your suggestion and comment. Rephrased as: "The importance of $g_m$ in constraining $A$ was variable, and depended on leaf structural traits, only LMA, LT, and LD were analyzed in this study."

*(33)Lines 368-374: chloroplasts do not have cell walls, the sentences need to berephrased*

Response: Thank you for your suggestion and comment. This mistake has been corrected.

*(34) Line 402: "highly efficient"*

Response: This change has been made.

*(35)Line 411: delete the first "in this study"*

Response: Deleted.

*(36)Line 415: "lose" not "loss"*

Response: Corrected. Thank you.

*(37) Lines 416-417 "The results ...": unnecessary sentence, delete*

Response: Deleted.

*(38)Line 422: full stop missing from the end*

Response: Added. Thank you.

*(39) Line 424: delete "The"*

Response: This change has been made.

*(40) Lines 424-425 stating with "In theory": should be in the present*

Response: This change has been made.

*(41) Line 433: This sentence should be in the present*

This change has been made.

*(42) Line 448: ...inefficiency in "the" trade-off*

Response: This change has been made.

*(43) Line 452: "low nutrient"*

Response: This change has been made.

*(44) Line 461: iWUE is not in italic in any other place*

Response: This change has been made.

*(45)Line 462: ...forms in "the" field*

Response: This change has been made.

*(46) Line 463: ... used "a" diverse*

Response: Change has been made, thank you.

*(47) Line 464: ... maintain "a" relatively*

Response: This change has been made.

*(48) Line 465: ... used "the"*

Response: Thank you for your suggestion and comment. Chang has been made.

*(49)Line 483: "References"*

Response: Change has been made.

**Appendix**

**1 Figures**

[revised manuscript text omitted]

Figure S4 Relationship between (a) light-saturated net photosynthesis (*A*) and the leaf thickness (LT); (b) *A* and he leaf density (LD); (c) the ratio of *A* to leaf mass per area (LMA) (*A*/LMA); and (d) *A*/LMA and LD. Lines refer to regression line for 63 species. T, TS, S, G, V, and F represent Tree, Tree/Shrub, Shrub, Grass, Vine, and Fern, respectively.

[Figure]

Figure S5 Relationship between (a) the mesophyll conductance to $CO_2$ ($g_m$) and the leaf thickness (LT); (b) $g_m$ and he leaf density (LD); (c) the ratio of $g_m$ to leaf mass per area (LMA) ($g_m$/LMA); and (d) $g_m$/LMA and LD. Lines refer to regression line for 63 species. T, TS, S, G, V, and F represent Tree, Tree/Shrub, Shrub, Grass, Vine, and Fern, respectively.

[Figure]

Figure S6 Relationship between (a) the maximum carboxylase activity of Rubisco ($V_{cmax}$) and the leaf thickness (LT); (b) $V_{cmax}$ and he leaf density (LD); (c) the ratio of $V_{cmax}$ to leaf mass per area (LMA) ($V_{cmax}$/LMA); and (d) $V_{cmax}$/LMA and LD. Lines refer to regression line for 63 species. T, TS, S, G, V, and F represent Tree, Tree/Shrub, Shrub, Grass, Vine, and Fern, respectively.

**2 Tables**

Table S1 Details information about the 63 species in the subtropical primary forest in Southwest China.

| Species | Plant family | | Life form | |
|---|---|---|---|---|
| Broussonetia papyifera (Linn.) L'Hert. ex Vent. | Moraceae | Tree | Deciduous | Woody |
| Machilus microcarpa Hemsl. | Lauraceae | Tree | Evergreen | Woody |
| Melia azedarach L. | Meliaceae | Tree | Deciduous | Woody |
| Populus ×canadensis Moench. | Salicaceae | Tree | Deciduous | Woody |
| Camptotheca acuminata Decne. | Nyssaceae | Tree | Deciduous | Woody |
| Cinnamomum bodinieri Levl. | Lauraceae | Tree | Evergreen | Woody |
| Catalpa ovata G. Don | Bignoniaceae | Tree | Deciduous | Woody |
| Toona sinensis (A. Juss.) Roem. | Meliaceae | Tree | Deciduous | Woody |
| Sapium sebiferum (Linn.) Roxb. | Euphorbiaceae | Tree | Deciduous | Woody |
| Cladrastis platycarpa (Maxim.) Makino | Leguminosae | Tree | Deciduous | Woody |
| Ulmus pumila L. | Ulmaceae | Tree | Deciduous | Woody |
| Ilex macrocarpa Oliv. | Aquifoliaceae | Tree | Deciduous | Woody |
| Vitex canescens Kurz | Verbenaceae | Tree | Deciduous | Woody |
| Eriobotrya japonica (Thunb.) Lindl. | Rosaceae | Tree | Evergreen | Woody |
| Morus alba L. | Moraceae | Tree | Deciduous | Woody |
| Prunus salicina Lindl. | Rosaceae | Tree | Deciduous | Woody |
| Eucommia ulmoides Oliver | Eucommiaceae | Tree | Deciduous | Woody |
| Platycarya strobilacea Sieb. et Zucc. | Juglandaceae | Tree | Deciduous | Woody |
| Kalopanax septemlobus (Thunb.) Koidz. | Araliaceae | Tree | Deciduous | Woody |
| Zanthoxylum armatum DC. | Rutaceae | Tree | Deciduous | Woody |
| Pyrus calleryana | Rosaceae | Tree | Deciduous | Woody |
| Amygdalus persica L. var. | Rosaceae | Tree | Deciduous | Woody |

| Species | Family | Habit | Leaf | Type |
|---|---|---|---|---|
| Euonymus meaackii Rupr. | Celastraceae | Tree | Deciduous | Woody |
| Zanthoxylum ovalifolium Wight | Rutaceae | Tree | Deciduous | Woody |
| Cerasus scopulorum (Koehne) Yu et Li | Rosaceae | Tree | Deciduous | Woody |
| Carpinus pubescens Burk. | Betulaceae | Tree | Deciduous | Woody |
| Lithocarpus confinis Huang | Fagaceae | Tree | Evergreen | Woody |
| Celtis sinensis Pers. | Ulmaceae | Tree | Deciduous | Woody |
| Diospyros kaki Thunb. var. silvestris Makino | Ebenaceae | Tree | Deciduous | Woody |
| Ligustrum lucidum Ait. | Oleaceae | Tree/Shrub | Deciduous | Woody |
| Rhamnus leptophylla Schneid. | Rhamnaceae | Tree/Shrub | Deciduous | Woody |
| Lindera communis Hemsl. | Lauraceae | Tree/Shrub | Evergreen | Woody |
| Itea yunnanensis Franch | Saxifragaceae | Tree/Shrub | Evergreen | Woody |
| Pittosporum brevicalyx (Oliv.) Gagnep | Pittosporaceae | Tree/Shrub | Evergreen | Woody |
| Litsea rubescens Lec. | Lauraceae | Tree/Shrub | Deciduous | Woody |
| Rhus chinensis Mill. | Anacardiaceae | Tree/Shrub | Deciduous | Woody |
| Alangium chinense (Lour.) Harms | Alangiaceae | Tree/Shrub | Deciduous | Woody |
| Evodia rutaecarpa (Juss.) Benth. | Rutaceae | Tree/Shrub | Deciduous | Woody |
| Machilus cavaleriei Levl. | Lauraceae | Tree/Shrub | Evergreen | Woody |
| Debregeasia longifolia (Burm. f.) Wedd. | Urticaceae | Tree/Shrub | Deciduous | Woody |
| Ziziphus jujuba Mill. var. spinosa (Bunge) Hu ex H. F. Chow | Rhamnaceae | Shrub | Deciduous | Woody |
| Rubus inopertus (Diels) Focke | Rosaceae | Shrub | Deciduous | Woody |
| Coriaria nepalensis Wall. | Coriariaceae | Shrub | Deciduous | Woody |
| Celastrus orbiculatus Thunb. | Celastraceae | Shrub | Deciduous | Woody |
| Wikstroemia scytophylla Diels | Thymelaeaceae | Shrub | Deciduous | Woody |
| Viburnum foetidum Wall. var. ceanothoides (C. H. Wright) Hand.-Mazz. | Caprifoliaceae | Shrub | Deciduous | Woody |
| Hedera nepalensis K. Koch var. sinensis (Tobl.) Rehd. | Araliaceae | Shrub | Deciduous | Woody |
| Rubus parvifolius L. | Rosaceae | Shrub | Deciduous | Woody |

| | | | | |
|---|---|---|---|---|
| Rosa roxbunghii | Rosaceae | Shrub | Deciduous | Woody |
| Mallotus repandus (Willd.) Muell. Arg. | Euphorbiaceae | Shrub | Deciduous | Woody |
| Mahonia bealei (Fort.) Carr. | Berberidaceae | Shrub | Evergreen | Woody |
| Fallopia multiflora (Thunb.) Harald. | Polygonaceae | Grass | | Herb |
| Conyza canadensis (L.) Cronq. | Compositae | Grass | | Herb |
| Ipomoea batatas (L.) Lam. | Convolvulaceae | Grass | | Herb |
| Senecio scandens Buch.-Ham. ex D. Don | Compositae | Grass | | Herb |
| Vitis piasezkii Maxim. | Vitaceae | Vien | Deciduous | Woody |
| Clematis urophylla Franch. | Ranunculaceae | Vien | Deciduous | Woody |
| Bauhinia glauca (Wall. ex Benth.) Benth. | Leguminosae | Vien | Evergreen | Woody |
| Caesalpinia decapetala (Roth) Alston | Leguminosae | Vien | Deciduous | Woody |
| Paederia scandens (Lour.) Merr. | Rubiaceae | Vien | | Herb |
| Cyclosorus parasiticus (L.) Farwell. | Thelypteridaceae | Fern | | |
| Cyrtomium fortunei J. Sm. | Dryopteridaceae | Fern | | |
| Pteris vittata L. | Pteridaceae | Fern | | |

Table S2 Coefficients of determination of linear regressions of fig. 1-4 and fig.6-7.

| Subgraph | Life form | Fig.1 | | Fig.2 | | Fig.3 | | Fig.4 | | Fig.6 | | Fig.7 | |
|---|---|---|---|---|---|---|---|---|---|---|---|---|---|
| | | $R^2$ | P | $R^2$ | P | $R^2$ | P | $R^2$ | P | $R^2$ | P | $R^2$ | P |
| a | Total | 0.35 | 0.000 | 0.09 | 0.018 | 0.67 | 0.000 | 0.19 | 0.000 | 0.00 | 0.922 | 0.20 | 0.000 |
| | Tree | 0.49 | 0.000 | 0.14 | 0.048 | 0.67 | 0.000 | 0.42 | 0.000 | 0.03 | 0.401 | 0.11 | 0.083 |
| | Tree/Shrub | 0.70 | 0.001 | 0.49 | 0.016 | 0.79 | 0.000 | 0.57 | 0.007 | 0.24 | 0.126 | 0.07 | 0.438 |
| | Shrub | 0.29 | 0.085 | 0.10 | 0.350 | 0.78 | 0.000 | 0.11 | 0.314 | 0.00 | 1.000 | 0.20 | 0.173 |
| b | Total | 0.75 | 0.000 | 0.47 | 0.000 | 0.53 | 0.000 | 0.65 | 0.000 | 0.34 | 0.000 | 0.52 | 0.000 |
| | Tree | 0.85 | 0.000 | 0.53 | 0.000 | 0.42 | 0.000 | 0.80 | 0.000 | 0.49 | 0.000 | 0.58 | 0.000 |
| | Tree/Shrub | 0.84 | 0.000 | 0.67 | 0.002 | 0.68 | 0.002 | 0.78 | 0.000 | 0.70 | 0.001 | 0.78 | 0.000 |
| | Shrub | 0.60 | 0.005 | 0.50 | 0.015 | 0.75 | 0.001 | 0.42 | 0.031 | 0.22 | 0.142 | 0.56 | 0.008 |
| c | Total | 0.55 | 0.000 | 0.59 | 0.000 | 0.76 | 0.000 | 0.38 | 0.000 | 0.00 | 0.934 | | |
| | Tree | 0.68 | 0.000 | 0.67 | 0.000 | 0.70 | 0.000 | 0.63 | 0.000 | 0.01 | 0.549 | | |
| | Tree/Shrub | 0.79 | 0.000 | 0.88 | 0.000 | 0.83 | 0.000 | 0.67 | 0.002 | 0.21 | 0.162 | | |
| | Shrub | 0.50 | 0.014 | 0.55 | 0.009 | 0.84 | 0.000 | 0.23 | 0.138 | 0.01 | 0.771 | | |
| d | Total | 0.25 | 0.000 | | | 0.22 | 0.000 | 0.27 | 0.000 | 0.09 | 0.016 | | |
| | Tree | 0.36 | 0.001 | | | 0.09 | 0.121 | 0.34 | 0.001 | 0.08 | 0.133 | | |
| | Tree/Shrub | 0.40 | 0.038 | | | 0.02 | 0.714 | 0.52 | 0.013 | 0.19 | 0.180 | | |
| | Shrub | 0.04 | 0.552 | | | 0.53 | 0.011 | 0.01 | 0.734 | 0.06 | 0.471 | | |

---

## Author Comment (AC2) · 25 Apr 2018

Response to reviews of manuscript "Trade-offs between water loss and carbon gain in a subtropical primary forest on Karst soils in China" bg-2018-44

Response to reviewer#2

Dear Reviewer,

We would like to thank you for the thoughtful and valuable comments and suggestions on our manuscript entitled "Trade-offs between water loss and carbon gain in a subtropical primary forest on Karst soils in China" (bg-2018-44). We have carefully revised our manuscript to take account of your comments and suggestions. Please find below our responses (upright Roman) to comments (original queries in Italic). Meanwhile,

we have rephrased our manuscript title as "The strategies of water-carbon regulation of plants in a subtropical primary forest on Karst soils in China". The line numbers mentioned here refer to our original manuscript. The changed figures and tables are presented in the Appendix (listed at the end of the "Response to reviewer").

General comments:

(1)ãÁÁThe author use "Trade-offs between water loss and carbon gain" in the title, however, the whole-text actually talk about the limitation of different components on A and iWUE.

Response: Thanks a lot for your comment. We response to this comment from two aspects. One on hand, we have rephrased our manuscript title as "The strategies of water-carbon regulation of plants in a subtropical primary forest on Karst soils in China".

On the other hand, we have revised the Section "Discussion". Firstly, we have re-organized and revised Section "4.1 The role of gm in $CO_2$ diffusion and Vcmax", and merged it with "4.2 Co-variation in gs, gm and Vcmax in regulating A ". Such as, we have moved two paragraphs "Three methods are most commonly used for gm estimation. ..... All of these methods are based on gas exchange measurements (Pons et al., 2009), and some common assumptions (Warren, 2006). Thus, the accuracy of each method is largely unknown (Warren, 2006) (Lines 288-295). The gm was estimated by the 'curve-fitting' method in this study. ... We confirmed that the calculated Cc and the initial slope of A-Cc curves were positive, suggesting that the measured gm was reliable (Warren, 2006). (Lines 297-304)" to Section "Methods and Materials".

We have deleted two paragraphs "Large uncertainties can be introduced by ignoring gm. .... âŰş13C_gm represented the carbon isotope discrimination when gm was finite, and âŰş13C_gs represented the carbon isotope discrimination when gm was infinite (Lines 319-328). On the other hand, ignoring gm would underestimate Vcmax up to 75% (Sun et al., 2014). .... Furthermore, the leaf barrier to $CO_2$ caused by gm has

not been represented in the global carbon cycles, leading to an overestimation of CO2 supply for carboxylation and an underestimation of the response of photosynthesis to atmospheric CO2 (Sun et al., 2014) (Lines 330-337).".

We have revised the paragraph "Large variability in gm has been shown both between and within species with different leaf forms and habits (Gago et al., 2014; Flexas et al., 2016). .... Hence, the wide variability of gm between different species and life forms in the same ecosystem seems to be related to the diversity in leaf anatomical traits. (Lines 306-317)" to "The importance of gm in constraining A was variable, and depended on leaf structural traits, only LMA, LT, and LD were analyzed in this study. Large variability in gm has been shown both between and within species with different life forms and habits (Gago et al., 2014; Flexas et al., 2016). Variability in gm in this study is similar to that in global datasets (Gago et al., 2014; Flexas et al., 2016). There was no significantly difference among life forms (P>0.05). Previous studies have confirmed that LMA (Tomas et al., 2013), thickness of leaf cell wall (Peguero-Pina et al., 2017b), liquid phase of mesophyll (Veromann-Jurgenson et al., 2017), cell wall thickness of mesophyll (Terashima et al., 2011;Tosens et al., 2016), and surface area of mesophyll and chloroplast exposed to intercellular space (Veromann-Jurgenson et al., 2017) were the main limitations for gm. The wide variability of gm between different species and life forms in the same ecosystem seems to be related to the diversity of leaf anatomical traits.". And we have merged this paragraph with "4.2 Co-variation in gs, gm and Vcmax in regulating A ".

Secondly, we revised the title of Section "4.2 Co-variation in gs, gm and Vcmax in regulating A" as "4.1 Co-variation in gs, gm and Vcmax in regulating A ". And we have reanalyzed our data, and revised the paragraph "The A was constrained by gs, gm, and Vcmax acting together, however, variability in the relative contribution of these three factors depended on species and habitats (Tosens et al., 2016; Galmes et al., 2017; Peguero-Pina et al., 2017a; Veromann-Jurgenson et al., 2017)..... In addition, 20 of the 63 species were mainly limited by Vcmax (lb>0.4, with the largest value of 0.68).

(Lines 340-351) " to "A was constrained by gs, gm, and Vcmax acting together, however, variability in the relative contribution of these three factors depended on species and habitats (Tosens et al., 2016; Galmes et al., 2017; Peguero-Pina et al., 2017a; Veromann-Jurgenson et al., 2017). A was significantly correlated with gs, gm, and Vcmax (Fig.3a-c). gs was positively related to gm (Fig.S1c), while no relationship was found between the $CO_2$ diffusion conductance (gs and gm) and Vcmax (Fig. S2). The relative limitations of gs, gm, and Vcmax were separated by a quantitative limitation model (Jones, 1985; Grassi & Magnani, 2005). The results showed that ls, lm and lb of 63 species varied in a large range (Fig. S3), indicating plants have a diverse strategies to co-ordinate the $CO_2$ diffusion (gs and gm) and Vcmax to maintain relative high A. The order of factors limitations to A was lm> lb >ls ($P<0.05$) (Fig.S3). Furthermore, we tested the relationship between the relative limitations and the corresponding limitation factors. The results showed that ls, lm, and lb were negatively associated with gs, gm, and Vcmax, respectively (Fig. 4). And the relationship was stronger for gm- lm ($r2=0.65$) than Vcmax- lb ($r2=0.27$) and gs- ls ($r2=0.19$).

gs was better correlated with A, while the results showed that A was more limited by gm. That could be explained by two possible reasons. Firstly, compare to the linear relationship between A and gs, a nonlinear trend has been found between A and gm when gm>0.4 (Fig. 3a, b). Secondly, leaf structure plays an important role in regulating gm and Vcmax, consequently, in determining A (Veromann-Jurgenson et al., 2017). Negative relationships between A/LMA and LT ($r2=0.16$, $p=0.002$), and A/LMA and LT ($r2=0.3$, $p<0.001$) have been observed (Fig. S4c,d), while A was not correlated to LT and LD (Fig. S4a,b)."

We have tested the difference of LMA, leaf thickness (LT) and leaf density (LD) among life forms, no significantly different have been found. And then we tested the roles of leaf structure (LT and LD) on A, gm, and Vcmax. The results showed that leaf structure plays important role in regulating A, gm, and Vcmax. Consequently, we revised discussions about the carbon fixation (A) strategies of plants (Lines 353-406) as

"The importance of gm in constraining A was variable, and depended on leaf structural traits, only LMA, LT, and LD were analyzed in this study. Large variability in gm has been shown both between and within species with different life forms and habits (Gago et al., 2014; Flexas et al., 2016). Variability in gm in this study is similar to that in global datasets (Gago et al., 2014; Flexas et al., 2016). There was no significantly difference among life forms (P>0.05). Previous studies have confirmed that LMA (Tomas et al., 2013), thickness of leaf cell wall (Peguero-Pina et al., 2017b), liquid phase of mesophyll (Veromann-Jurgenson et al., 2017), cell wall thickness of mesophyll (Terashima et al., 2011;Tosens et al., 2016), and surface area of mesophyll and chloroplast exposed to intercellular space (Veromann-Jurgenson et al., 2017) were the main limitations for gm. The wide variability of gm between different species and life forms in the same ecosystem seems to be related to the diversity of leaf anatomical traits.

No significant difference of LMA, LT, and LD was found among life forms (P<0.05). The negative correlation of gm (Terashima et al., 2005) or gm/LMA (Niinemets et al., 2009; Veromann-Jurgenson et al., 2017) with LMA have been reported. In this study, there was a significant relationship between gm/LMA with LMA (P<0.01), however, no relationship was found between gm with LMA. gm/LMA was significantly negative related to LD (p<0.01) (Fig. S5c), and weak negative related to LT (p=0.06) (Fig. S5d), demonstrating that the negative role of cell wall thickness on gm (Terashima et al., 2006; Niinemets et al., 2009). The strong investment in supportive structures was the main reason for the limitation of gm on A (Veromann-Jurgenson et al., 2017). However, it is still unknown how leaf anatomical traits affect gm and A, and this should be further explored.

gs is responsible for CO2 exchange between atmosphere and leaf, and regulate the CO2 fixation (A) and water loss (Lawsonand Blatt, 2014). The variability of gs was controlled by stomatal anatomy, i.e. stomata density and size, and mesophyll demands for CO2 (Lawsonand Blatt, 2014). However, the stomatal anatomy was not analyzed in this study. We only focused on how the relationship between gs and gm regulate A.

Positive relationship between gs and gm has been observed (Flexas et al., 2013). For example. the restricted CO2 diffusion from the ambient air to chloroplast is the main reason for a decreased A under water stress conditions due to both the stomatal and mesophyll limitations (Olsovska et al., 2016). gs was significantly positive related to gm for 63 species (P<0.001, Fig. S1) in this study, and no difference of the slopes of regression lines between gs and gm was found among life forms, demonstrating that A was regulated by the co-variation of gs and gm. However, the variability of gm and lm was larger than gs and ls, respectively (Fig.1 and Fig.S3).

The wide variation range of lb (0.11-0.68) highlighted the importance role of Vcmax in regulating A. Vcmax was used to represent the CO2 demand in photosynthetic process in this study. The relative contribution of Vcmax to A not only depends on Ca-Cc, but also on leaf nutrient levels. Positive relationship was found between Ca-Cc and Vcmax (Fig. 1d). And the Vcmax/LMA was co-regulated by leaf N, P and Mg content (Jing et al. 2018). In addition, Vcmax/LMA was negatively related to LT (p<0.05) (Fig. S6c) and LD (p<0.05) (Fig. S6d), while Vcmax was not correlated to LT and LD (Fig. S6a,b), demonstrating that leaf structure plays an important role in regulating Vcmax.

The trade-off between CO2 supply (gs and gm) and demand (carboxylation capacity of Rubisco) can help maintain relative high A (Galmes et al., 2017; Saez et al., 2017). In this study, we used Vcmax as a proxy for the carboxylation capacity of Rubisco, and the normalized Vcmax by A (V=Vcmax/A) was significantly negatively correlated with the normalized gt by A (Gt =gt/A) (P<0.001) (Fig. 2c), indicating that the trade-off between CO2 supply and demand also existed among different species in the same ecosystems. For genus Limonium (flowering plants) (Galmes et al., 2017), gt was significantly positively related to Rubisco carboxylase specific activity, and significantly negatively related to Rubisco specificity factor to CO2. In case of Antarctic vascular (Saez et al., 2017) and Mediterranean plants (Flexas et al., 2014), A was mainly limited by low gm, but it could be partially counterbalanced by a highly efficient Rubisco through high specificity for CO2. This highlights the importance of the trade-off between CO2 supply

and demand in plant adaptation to Karst environment. However, it is still unknown how leaf anatomical traits affect gm, Vcmax and A, and this should be further explored. "

Thirdly, we have revised the title of Section "4.3 Co-variation in gs, gm and Vcmax in regulating iWUE" as "4.2 Co-variation in gs, gm and Vcmax in regulating iWUE ". To emphasize the diverse carbon-water regulation strategies of plants in Karst CZs, and highlighted the role of trade-off between carbon gain and water loss, we have revised the paragraph "Compared with the global dataset under well-watered conditions (19.27-171.88 $\mu$mol $CO_2$ mol-1 H2O) (Flexas et al., 2016), the iWUE (29.52-88.92 $\mu$mol $CO_2$ mol-1 H2O) in this study was somewhat lower in this study....The average iWUE of 12 vines and 13 trees in the Karst tropical primary forest was 41.23±13.21 $\mu$mol $CO_2$ mol-1 H2O (Chen et al., 2015), while that of 6 evergreen and 6 deciduous trees was 66.7±4.9 and 49.7±2.0 $\mu$mol $CO_2$ mol-1 H2O, respectively (Fu et al., 2012). (Lines 409-422)" to "Compared with the global dataset under well-watered conditions (19.27-171.88 $\mu$mol $CO_2$ mol-1 H2O) (Flexas et al., 2016), iWUE (52.85±13.08 $\mu$mol $CO_2$ mol-1 H2O) was somewhat lower in this study. iWUE varied from 29.53 to 88.91 $\mu$mol $CO_2$ mol-1 H2O, and the variability of iWUE was larger than in the Karst tropical primary forest (Fu et al., 2012; Chen et al., 2015). The average iWUE of 12 Vines and 13 Trees in the Karst tropical primary forest was 41.23±13.21 $\mu$mol $CO_2$ mol-1 H2O (Chen et al., 2015), while that of 6 evergreen and 6 deciduous Trees was 66.7±4.9 and 49.7±2.0 $\mu$mol $CO_2$ mol-1 H2O, respectively (Fu et al., 2012). The results demonstrated that Karst plants use a diverse strategies of carbon-water regulation to adapt to the harsh Karst environment.

The trade-off between carbon gain and water loss is one of important strategies of carbon-water regulation of plants, and was exist among species and life forms (Prentice et al., 2014). Prentice et al. (2014) studied the trade-off between carbon gain and water loss of woody species in contrasting climates, and found that species in hot and wet regions tend to lose more water in order to fix more carbon (high gs/A, low Vcmax_Ci/A), and vice versa. Although Karst soils cannot contain enough water for

plant growth, the trade-off between carbon gain and water loss (high gs/A and low Vc-max_Ci/A) were similar to the shown for plants growing in hot and wet regions (Prentice et al., 2014). "

(2) In the method section: The species covered wide range of functional groups, including 6 life forms. What the criteria of the species selection? Because the leaf habit (evergreen or deciduous), the shade or light-demanding behaviors also will affect the strategy of plant carbon-water regulation. For example, does fern grow in the canopy or understory, how you can put them together when analyze the data? More important, the main objective of this paper was to determine and distinguish the limitations of CO2 diffusion and Vcmax on A and iWUE in different life forms Karst forest, however, you combine all species together for most analysis, actually we donot know what's the difference between different life forms in Figs 1-4, 6,7. I Believe most land plant will behave in similar way to adapt to the environmental factor no matter where they grow, the interesting things is to what extent by different plants. For example, Based on Fig 5, we could not see any difference among the groups. So, I suggest the author should separate into 6 groups to see the differences of regression lines among groups for all the figures, and compare the difference among the life forms using proper statistical method.

Response: Thank you for your comments and suggestions. We response to revised the manuscript from three aspects according to your comments and suggestions. Firstly, we have added our criteria of the species selection in Section "2.2 Leaf gas-exchange measurements" "In July and August 2016, 63 species (Table S1) were selected for measurements of the A and CO2 response curves. The species sampled were selected according to their abundance in the study site. They are the main component of this forest, including 55 woody species (46 deciduous and 10 evergreen species) and 5 herb species. To distinguish the strategies of water-carbon regulation of plants among different life forms, those species were grouped into 6 life forms, including (1) Tree (n=29), (2) Tree/Shrub (n=11), (3) Shrub (n=11), (4) Grass (n=11), (5) Vine (n=5),

and (6) Fern (n=3). "Tree/Shrub" is a kind of low wood plant between Tree and Shrub. Fern grow in understory. Vine climb up to the shrub canopy to get light. " We have added how were the leaves collected "Branches exposed to the sun were excised from the upper part of the crown (Trees, Tree/Shrubs, Shrubs and Vines) or aboveground portion (Grasses, Ferns), and immediately re-cut under water to maintain xylem water continuity. ".

Secondly, we have re-analyzed our data either as a whole group (six life forms combined) or by individual life forms, and the difference between different life forms was tested using the standardized major axis (SMA) regression fits. The results showed that no significantly difference between life forms. Thus six life forms were grouped together to analyze the strategy of water-carbon regulation of plants in the whole text. The statistical method and results have been added in Section "2.5 Statistical analysis" "Data were analyzed either as a whole group (six life forms combined) or by individual life forms. The bivariate linear regressions of leaf gas exchange parameters were performed using the standardized major axis (SMA) regression fits, and all of the data were made on loge-transformed data (Table S2).

To test for the differences among life forms, SMA regression fits were used to compare the slope of regression lines which significant relationships had already been obtained. Note that Grass, Vine and Fern were not considered due to the small sample size. A similar trend was obtained, and no significant difference was found between life forms although significant relationships were not obtained for some bivariate linear regressions. Accordingly, six life forms were grouped together to analyze the strategy of water-carbon regulation of plants in the whole text.

The difference of relative limitation of gs, gm and Vcmax to A for life forms or as a whole group were performed using one-way ANOVA and Duncan multiple comparison. The probability of significance was defined at p< 0.05. "

Thirdly, all of data of six life forms were separately presented in Figure 1-4, 6,7 and

[Figure]

Figure S1,S2, S4-S6 (See Appendix). Only the regression line for 63 species were presented in figures.

(3) lines 139-140, because the A-Ci curve is the key data of this paper, author should describe in detail how this measurement was done rather than just cite other submitted papers. For example, you should introduce the height of your targeted individuals? how you can measure the sun-exposed leaf for canopy trees and climbing plants: : :.?did you measure in situ or cut down, if the latter, for A-Ci curve you normally need ca. 30 min, how you can avoid the effects of cutting on stomatal conductance because some species are very sensitive, do you have some information on the gs sensitivity for those speciesïij§: : :..

Response: Thank you for your suggestions. In response, we have added more details about leaf sampling and measurements in Section "Materials and Methods". Such as, we have added the method of how were the leaves collected and prepared before $CO_2$ response curves measurements "Details of leaf sampling and measurements of the $CO_2$ response curve were briefly described as follows. Branches exposed to the sun were excised from the upper part of the crown (Trees, Tree/Shrubs, Shrubs and Vines) or above the ground (Grasses, Ferns), and immediately re-cut under water to maintain xylem water continuity. Back into the laboratory, branches were kept at 25oC for 30 min. Fully-expanded and mature leaves were induced for 30 minutes at a saturating light density (1500 $\mu$mol m-2 s-1). $CO_2$ response curves measurements were performed when A and gs was stable. Three leaves per species were collected and measured. A total of 189 leaves were collected from adult individuals of 63 species." However, the height of targeted individuals did not measured.

We have described the method and conditions of $CO_2$ response curves measurements in more detail as: "The $CO_2$ response curves were measured with 11 $CO_2$ concentration gradients in chamber following the procedural guidelines described by Longand Bernacchi (2003). The photosynthetic photon flux density was 1500 $\mu$mol m–2 s–1. The leaf temperature was 25°C, controlled by the block temperature. The humidity in

the leaf chamber was maintained at ambient condition. Leaf area, thickness (LT) and dry mass were measured after the CO2 response measurements. Leaf mass per area (LMA) was calculated by dividing the corresponding dry mass by leaf area. And leaf density (LD) was calculated by dividing the corresponding LMA by LT. More details were described in Wang et al. (2018)."

Specific comments:

(4) Line 267-269: There is no statistic tests of the differences of the results in figure 5, so it is not proper to give the statements in line 309-310. Figure 5 can't give any information that is about LMA. Please use data to demonstrate the relationship between LMA and other parameters instead of qualitative description.

Response: Thank you for your comments and suggestions. We response to the comments and suggestions from two aspects. Firstly, we have analyzed the data of figure 5 using statistical method, and revised the corresponding Sections. Such as, we have added statistical method used to test the difference of the results in figure 5 in Section "2.5 Statistical analysis" "The difference of relative limitation of gs, gm and Vcmax to A for life forms or as a whole group were performed using one-way ANOVA and Duncan multiple comparison. The probability of significance was defined at p< 0.05." .

We have drew figure 5 and revised the Section "3.2 Contribution of gs, gm and Vcmax to A" as "The variation in A was attributed to variation in gs, gm, gt, and Vcmax. A was positively correlated with gs (Fig. 3a), gm (Fig. 3b), and Vcmax (Fig. 3c). We used the quantitative limitation model (Eqs. (9), (10) and (11)) to separate gs (ls), gm (lm), and Vcmax (lb) limitations to A. ls, lm, and lb were negatively associated with gs, gm, and Vcmax, respectively (Fig. 4). The contributions by gs, gm, and Vcmax to limiting A were different for each species (Fig. S3). ls varied 2.6-fold ( from 0.17 to 0.45), lm varied 10.5-fold ( from 0.05 to 0.55), and lb varied 6.2-fold ( from 0.11 to 0.68) across species. Overall, lm (0.38±0.12) was significantly larger than lb (0.34±0.14), and ls (0.28±0.07) (P<0.05).

[Figure]

To further understand how A was limited by gs, gm, and Vcmax among life forms, we grouped the 63 species into 6 life forms: Tree, Tree/Shrub, Shrub, Grass, Vine, and Fern. The results showed that there was no significantly difference between ls, lm and lb for Trees and Tree/shrubs. lm of Shrubs and Grasses was significantly higher than that of ls and lb (P<0.05). lm of Vines and Ferns was significantly higher than that of ls (P<0.05) (Fig. 5). ". We have revised the Section "4.1 Co-variation in gs, gm and Vcmax in regulating A ". Please also see the response to reviewer #1.

Secondly, we have tested the difference of LMA across life forms using one-way ANOVA and Duncan multiple comparison. The results showed that no difference of LMA was found among life forms. Consequently, lines 309-310 have been removed. We have tested the role of leaf structure (leaf thickness (LT) and leaf density (LD)) in A, gm and Vcmax, and rephrased the Section "4.1 Co-variation in gs, gm and Vcmax in regulating A ". Please also see the response to reviewer #1.

(5) Line 372: Species with low LMA may have thick cell walls in mesophyll and chloroplast.

Response: Thank you for your suggestion. We have tested the difference of LMA across life forms using one-way ANOVA and Duncan multiple comparison. The results showed that no difference of LMA was found among life forms. Meanwhile, We have tested the role of leaf structure (leaf thickness (LT) and leaf density (LD)) in A, gm and Vcmax. The results showed that leaf structure plays important role in regulating gm and Vcmax, consequently, in determining A. Consequently, we revised the corresponding section in "4.1 Co-variation in gs, gm and Vcmax in regulating A " as "The importance of gm in constraining A was variable, and depended on leaf structural traits, only LMA, LT, and LD were analyzed in this study. Large variability in gm has been shown both between and within species with different life forms and habits (Gago et al., 2014; Flexas et al., 2016). Variability in gm in this study is similar to that in global datasets (Gago et al., 2014; Flexas et al., 2016). There was no significantly difference among life forms (P>0.05). Previous studies have confirmed that LMA (Tomas et al.,

2013), thickness of leaf cell wall (Peguero-Pina et al., 2017b), liquid phase of meso-phyll (Veromann-Jurgenson et al., 2017), cell wall thickness of mesophyll (Terashima et al., 2011;Tosens et al., 2016), and surface area of mesophyll and chloroplast exposed to intercellular space (Veromann-Jurgenson et al., 2017) were the main limitations for gm. The wide variability of gm between different species and life forms in the same ecosystem seems to be related to the diversity of leaf anatomical traits.

No significant difference of LMA, LT, and LD was found among life forms (P<0.05). The negative correlation of gm (Terashima et al., 2005) or gm/LMA (Niinemets et al., 2009; Veromann-Jurgenson et al., 2017) with LMA have been reported. In this study, there was a significant relationship between gm/LMA with LMA (P<0.01), however, no relationship was found between gm with LMA. gm/LMA was significantly negative related to LD (p<0.01) (Fig. S5c), and weak negative related to LT (p=0.06) (Fig. S5d), demonstrating that the negative role of cell wall thickness on gm (Terashima et al., 2006; Niinemets et al., 2009). The strong investment in supportive structures was the main reason for the limitation of gm on A (Veromann-Jurgenson et al., 2017). However, it is still unknown how leaf anatomical traits affect gm and A, and this should be further explored. "

(6) Line 381-382: In your results, gs and gm are positively correlated, why did you conclude gm is a compensate for reductions in gs? Did you observe an increasing of gm when gs decreased.

Response: Thank you for your comment. We corrected this mistake, and we rephrased this paragraph as: "gs is responsible for CO2 exchange between atmosphere and leaf, and regulate the CO2 fixation (A) and water loss (Lawsonand Blatt, 2014). The variability of gs was controlled by stomatal anatomy, i.e. stomata density and size, and mesophyll demands for CO2 (Lawsonand Blatt, 2014). However, the stomatal anatomy was not analyzed in this study. We only focused on how the relationship between gs and gm regulate A. Positive relationship between gs and gm has been observed (Flexas et al., 2013). For example. the restricted CO2 diffusion from the

ambient air to chloroplast is the main reason for a decreased A under water stress conditions due to both the stomatal and mesophyll limitations (Olsovska et al., 2016). gs was significantly positive related to gm for 63 species (P<0.001, Fig. S1) in this study, and no difference of the slopes of regression lines between gs and gm was found among life forms, demonstrating that A was regulated by the co-variation of gs and gm. However, the variability of gm and lm was larger than gs and ls, respectively (Fig.1 and Fig.S3)."

(7) Line 384-389: I don't think you have enough evidences to state "there was a trend of increasing lm with increasing leaf N:P", unless you add this part of research in your draft.

Response: Thank you for your comment. There was no significant statistical relationship between lm and leaf N:P (P=0.66). We corrected this mistake, and rephrased this paragraph : " The wide variation range of lb (0.11-0.68) highlighted the importance role of Vcmax in regulating A. Vcmax was used to represent the $CO_2$ demand in photosynthetic process in this study.The relative contribution of Vcmax to A not only depends on Ca-Cc, but also on leaf nutrient levels. Positive relationship was found between Ca-Cc and Vcmax (Fig. 1d). And the Vcmax/LMA was co-regulated by leaf N, P and Mg content (Jing et al. 2018). In addition, Vcmax/LMA was negatively related to LT (p<0.05) (Fig. S6c) and LD (p<0.05) (Fig. S6d), while Vcmax was not correlated to LT and LD (Fig. S6a,b), demonstrating that leaf structure plays an important role in regulating Vcmax."

(8) Awful sentences, Lines 39-35, should split into short sentences

Response: Rephrased as: "The results showed that (1) gs and gm varied about 7.6- and 34.5-fold, respectively, and gs was positively related to gm. The contribution of gm to leaf $CO_2$ gradient was similar to that of gs. The gs/A, gm/A and gt/A was negative related to Vcmax/A. (2) the relative limitations of gs (ls), gm (lm) and Vcmax (lb) to A for the whole group (combined 6 life forms) were significantly different from

each other (P<0.05). lm was the largest (0.38±0.12), followed by lb (0.34±0.14) and ls (0.28±0.07). No significant difference was found between ls, lm, and lb for Trees and Tree/shrubs, while lm was the largest, followed by lb and ls for Shrubs, Grasses, Viens and Ferns (P<0.05). (3) iWUE varied about 3-fold (from 29.52 to 88.92 $\mu$mol CO2 mol-1 H2O) across all species, and was significantly correlated with gs, Vcmax, gm/gs, and Vcmax/gs."

Please also note the supplement to this comment:
https://www.biogeosciences-discuss.net/bg-2018-44/bg-2018-44-AC2-supplement.pdf

**Supplement:**

Response to reviews of manuscript "Trade-offs between water loss and carbon gain in a subtropical primary forest on Karst soils in China" bg-2018-44

**Response to reviewer#2**

Dear Reviewer,

We would like to thank you for the thoughtful and valuable comments and suggestions on our manuscript entitled "Trade-offs between water loss and carbon gain in a subtropical primary forest on Karst soils in China" (bg-2018-44). We have carefully revised our manuscript to take account of your comments and suggestions. Please find below our responses (upright Roman) to comments (original queries in Italic). Meanwhile, we have rephrased our manuscript title as "The strategies of water-carbon regulation of plants in a subtropical primary forest on Karst soils in China". The line numbers mentioned here refer to our original manuscript. The changed figures and tables are presented in the Appendix (listed at the end of the "Response to reviewer").

General comments:

*(1)    The author use "Trade-offs between water loss and carbon gain" in the title, however, the whole-text actually talk about the limitation of different components on A and iWUE.*

Response: Thanks a lot for your comment. We response to this comment from two aspects. One on hand, we have rephrased our manuscript title as "The strategies of water-carbon regulation of plants in a subtropical primary forest on Karst soils in China".

On the other hand, we have revised the Section "Discussion". Firstly, we have re-organized and revised Section "4.1 The role of $g_m$ in $CO_2$ diffusion and $V_{cmax}$", and merged it with "4.2 Co-variation in $g_s$, $g_m$ and $V_{cmax}$ in regulating $A$ ". Such as, we have moved two paragraphs "Three methods are most commonly used for $g_m$ estimation. ..... All of these methods are based on gas exchange measurements (Pons

et al., 2009), and some common assumptions (Warren, 2006). Thus, the accuracy of each method is largely unknown (Warren, 2006) (Lines 288-295).

The $g_m$ was estimated by the 'curve-fitting' method in this study. ... We confirmed that the calculated $C_c$ and the initial slope of $A$-$C_c$ curves were positive, suggesting that the measured $g_m$ was reliable (Warren, 2006). (Lines 297-304)" to Section "Methods and Materials".

We have deleted two paragraphs "Large uncertainties can be introduced by ignoring $g_m$. .... $\Delta^{13}C\_g_m$ represented the carbon isotope discrimination when $g_m$ was finite, and $\Delta^{13}C\_g_s$ represented the carbon isotope discrimination when $g_m$ was infinite (Lines 319-328).

On the other hand, ignoring $g_m$ would underestimate $V_{cmax}$ up to 75% (Sun et al., 2014). .... Furthermore, the leaf barrier to $CO_2$ caused by $g_m$ has not been represented in the global carbon cycles, leading to an overestimation of $CO_2$ supply for carboxylation and an underestimation of the response of photosynthesis to atmospheric $CO_2$ (Sun et al., 2014) (Lines 330-337).".

We have revised the paragraph "Large variability in $g_m$ has been shown both between and within species with different leaf forms and habits (Gago et al., 2014; Flexas et al., 2016). .... Hence, the wide variability of $g_m$ between different species and life forms in the same ecosystem seems to be related to the diversity in leaf anatomical traits. (Lines 306-317)" to "The importance of $g_m$ in constraining $A$ was variable, and depended on leaf structural traits, only LMA, LT, and LD were analyzed in this study. Large variability in $g_m$ has been shown both between and within species with different life forms and habits (Gago et al., 2014; Flexas et al., 2016). Variability in $g_m$ in this study is similar to that in global datasets (Gago et al., 2014; Flexas et al., 2016). There was no significantly difference among life forms (P>0.05). Previous studies have confirmed that LMA (Tomas et al., 2013), thickness of leaf cell wall (Peguero-Pina et al., 2017b), liquid phase of mesophyll (Veromann-Jurgenson et al., 2017), cell wall thickness of mesophyll (Terashima et al., 2011;Tosens et al., 2016), and surface area

of mesophyll and chloroplast exposed to intercellular space (Veromann-Jurgenson et al., 2017) were the main limitations for $g_m$. The wide variability of $g_m$ between different species and life forms in the same ecosystem seems to be related to the diversity of leaf anatomical traits.". And we have merged this paragraph with "4.2 Co-variation in $g_s$, $g_m$ and $V_{cmax}$ in regulating $A$ ".

Secondly, we revised the title of Section "4.2 Co-variation in $g_s$, $g_m$ and $V_{cmax}$ in regulating $A$" as "4.1 Co-variation in $g_s$, $g_m$ and $V_{cmax}$ in regulating $A$ ". And we have re-analyzed our data, and revised the paragraph "The $A$ was constrained by $g_s$, $g_m$, and $V_{cmax}$ acting together, however, variability in the relative contribution of these three factors depended on species and habitats (Tosens et al., 2016; Galmes et al., 2017; Peguero-Pina et al., 2017a; Veromann-Jurgenson et al., 2017)..... In addition, 20 of the 63 species were mainly limited by $V_{cmax}$ ($l_b$>0.4, with the largest value of 0.68). (Lines 340-351) " to "$A$ was constrained by $g_s$, $g_m$, and $V_{cmax}$ acting together, however, variability in the relative contribution of these three factors depended on species and habitats (Tosens et al., 2016; Galmes et al., 2017; Peguero-Pina et al., 2017a; Veromann-Jurgenson et al., 2017). $A$ was significantly correlated with $g_s$, $g_m$, and $V_{cmax}$ (Fig.3a-c). $g_s$ was positively related to $g_m$ (Fig.S1c), while no relationship was found between the $CO_2$ diffusion conductance ($g_s$ and $g_m$) and $V_{cmax}$ (Fig. S2). The relative limitations of $g_s$, $g_m$, and $V_{cmax}$ were separated by a quantitative limitation model (Jones, 1985; Grassi & Magnani, 2005). The results showed that $l_s$, $l_m$ and $l_b$ of 63 species varied in a large range (Fig. S3), indicating plants have a diverse strategies to co-ordinate the $CO_2$ diffusion ($g_s$ and $g_m$) and $V_{cmax}$ to maintain relative high $A$. The order of factors limitations to $A$ was $l_m$> $l_b$ >$l_s$ (P<0.05) (Fig.S3). Furthermore, we tested the relationship between the relative limitations and the corresponding limitation factors. The results showed that $l_s$, $l_m$, and $l_b$ were negatively associated with $g_s$, $g_m$, and $V_{cmax}$, respectively (Fig. 4). And the relationship was stronger for $g_m$-$l_m$ (r$^2$=0.65) than $V_{cmax}$- $l_b$ (r$^2$=0.27) and $g_s$- $l_s$ (r$^2$=0.19).

[revised manuscript text omitted]

*(2) In the method section: The species covered wide range of functional groups, including 6 life forms. What the criteria of the species selection? Because the leaf habit (evergreen or deciduous), the shade or light-demanding behaviors also will affect the strategy of plant carbon-water regulation. For example, does fern grow in the canopy or understory, how you can put them together when analyze the data? More important, the main objective of this paper was to determine and distinguish the limitations of CO2 diffusion and Vcmax on A and iWUE in different life forms Karst forest, however, you combine all species together for most analysis, actually we donot know what's the difference between different life forms in Figs 1-4, 6,7. I Believe most land plant will behave in similar way to adapt to the environmental factor no matter where they grow, the interesting things is to what extent by different plants. For example, Based on Fig 5, we could not see any difference among the groups. So, I suggest the author should separate into 6 groups to see the differences of regression lines among groups for all the figures, and compare the difference among the life forms using proper statistical method.*

Response: Thank you for your comments and suggestions. We response to revised the manuscript from three aspects according to your comments and suggestions. Firstly,

we have added our criteria of the species selection in Section "2.2 Leaf gas-exchange measurements" "In July and August 2016, 63 species (Table S1) were selected for measurements of the $A$ and $CO_2$ response curves. The species sampled were selected according to their abundance in the study site. They are the main component of this forest, including 55 woody species (46 deciduous and 10 evergreen species) and 5 herb species. To distinguish the strategies of water-carbon regulation of plants among different life forms, those species were grouped into 6 life forms, including (1) Tree (n=29), (2) Tree/Shrub (n=11), (3) Shrub (n=11), (4) Grass (n=11), (5) Vine (n=5), and (6) Fern (n=3). "Tree/Shrub" is a kind of low wood plant between Tree and Shrub. Fern grow in understory. Vine climb up to the shrub canopy to get light. " We have added how were the leaves collected "Branches exposed to the sun were excised from the upper part of the crown (Trees, Tree/Shrubs, Shrubs and Vines) or aboveground portion (Grasses, Ferns), and immediately re-cut under water to maintain xylem water continuity. ".

Secondly, we have re-analyzed our data either as a whole group (six life forms combined) or by individual life forms, and the difference between different life forms was tested using the standardized major axis (SMA) regression fits. The results showed that no significantly difference between life forms. Thus six life forms were grouped together to analyze the strategy of water-carbon regulation of plants in the whole text. The statistical method and results have been added in Section "2.5 Statistical analysis" "Data were analyzed either as a whole group (six life forms combined) or by individual life forms. The bivariate linear regressions of leaf gas exchange parameters were performed using the standardized major axis (SMA) regression fits, and all of the data were made on $log_e$-transformed data (Table S2).

To test for the differences among life forms, SMA regression fits were used to compare the slope of regression lines which significant relationships had already been obtained. Note that Grass, Vine and Fern were not considered due to the small sample size. A similar trend was obtained, and no significant difference was found between life forms although significant relationships were not obtained for some bivariate

linear regressions. Accordingly, six life forms were grouped together to analyze the strategy of water-carbon regulation of plants in the whole text.

The difference of relative limitation of $g_s$, $g_m$ and $V_{cmax}$ to $A$ for life forms or as a whole group were performed using one-way ANOVA and Duncan multiple comparison. The probability of significance was defined at $p< 0.05$. ''

Thirdly, all of data of six life forms were separately presented in Figure 1-4, 6,7 and Figure S1,S2, S4-S6 (See Appendix). Only the regression line for 63 species were presented in figures.

*(3) lines 139-140, because the A-Ci curve is the key data of this paper, author should describe in detail how this measurement was done rather than just cite other submitted papers. For example, you should introduce the height of your targeted individuals? how you can measure the sun-exposed leaf for canopy trees and climbing plants: : :.?did you measure in situ or cut down, if the latter, for A-Ci curve you normally need ca. 30 min, how you can avoid the effects of cutting on stomatal conductance because some species are very sensitive, do you have some information on the gs sensitivity for those species ïj§: : :..*

Response: Thank you for your suggestions. In response, we have added more details about leaf sampling and measurements in Section "Materials and Methods". Such as, we have added the method of how were the leaves collected and prepared before $CO_2$ response curves measurements "Details of leaf sampling and measurements of the $CO_2$ response curve were briefly described as follows. Branches exposed to the sun were excised from the upper part of the crown (Trees, Tree/Shrubs, Shrubs and Vines) or above the ground (Grasses, Ferns), and immediately re-cut under water to maintain xylem water continuity. Back into the laboratory, branches were kept at 25$^{o}$C for 30 min. Fully-expanded and mature leaves were induced for 30 minutes at a saturating light density (1500 μmol m$^{-2}$ s$^{-1}$). $CO_2$ response curves measurements were performed when $A$ and $g_s$ was stable. Three leaves per species were collected and measured. A

total of 189 leaves were collected from adult individuals of 63 species." However, the height of targeted individuals did not measured.

We have described the method and conditions of $CO_2$ response curves measurements in more detail as: "The $CO_2$ response curves were measured with 11 $CO_2$ concentration gradients in chamber following the procedural guidelines described by Longand Bernacchi (2003). The photosynthetic photon flux density was 1500 μmol $m^{-2}$ $s^{-1}$. The leaf temperature was 25 ℃, controlled by the block temperature. The humidity in the leaf chamber was maintained at ambient condition. Leaf area, thickness (LT) and dry mass were measured after the $CO_2$ response measurements. Leaf mass per area (LMA) was calculated by dividing the corresponding dry mass by leaf area. And leaf density (LD) was calculated by dividing the corresponding LMA by LT. More details were described in Wang et al. (2018)."

Specific comments:

*(4) Line 267-269: There is no statistic tests of the differences of the results in figure 5, so it is not proper to give the statements in line 309-310. Figure 5 can't give any information that is about LMA. Please use data to demonstrate the relationship between LMA and other parameters instead of qualitative description.*

Response: Thank you for your comments and suggestions. We response to the comments and suggestions from two aspects. Firstly, we have analyzed the data of figure 5 using statistical method, and revised the corresponding Sections. Such as, we have added statistical method used to test the difference of the results in figure 5 in Section "2.5 Statistical analysis" "The difference of relative limitation of $g_s$, $g_m$ and $V_{cmax}$ to $A$ for life forms or as a whole group were performed using one-way ANOVA and Duncan multiple comparison. The probability of significance was defined at p< 0.05." .

We have drew figure 5 and revised the Section "3.2 Contribution of $g_s$, $g_m$ and $V_{cmax}$ to $A$" as "The variation in $A$ was attributed to variation in $g_s$, $g_m$, $g_t$, and $V_{cmax}$. $A$ was positively correlated with $g_s$ (Fig. 3a), $g_m$ (Fig. 3b), and $V_{cmax}$ (Fig. 3c). We used the

quantitative limitation model (Eqs. (9), (10) and (11)) to separate $g_s$ ($l_s$), $g_m$ ($l_m$), and $V_{cmax}$ ($l_b$) limitations to $A$. $l_s$, $l_m$, and $l_b$ were negatively associated with $g_s$, $g_m$, and $V_{cmax}$, respectively (Fig. 4). The contributions by $g_s$, $g_m$, and $V_{cmax}$ to limiting $A$ were different for each species (Fig. S3). $l_s$ varied 2.6-fold ( from 0.17 to 0.45), $l_m$ varied 10.5-fold ( from 0.05 to 0.55), and $l_b$ varied 6.2-fold ( from 0.11 to 0.68) across species. Overall, $l_m$ (0.38±0.12) was significantly larger than $l_b$ (0.34±0.14), and $l_s$ (0.28±0.07) (P<0.05).

To further understand how $A$ was limited by $g_s$, $g_m$, and $V_{cmax}$ among life forms, we grouped the 63 species into 6 life forms: Tree, Tree/Shrub, Shrub, Grass, Vine, and Fern. The results showed that there was no significantly difference between $l_s$, $l_m$ and $l_b$ for Trees and Tree/shrubs. $l_m$ of Shrubs and Grasses was significantly higher than that of $l_s$ and $l_b$ (P<0.05). $l_m$ of Vines and Ferns was significantly higher than that of $l_s$ (P<0.05) (Fig. 5). ”.

We have revised the Section "4.1 Co-variation in $g_s$, $g_m$ and $V_{cmax}$ in regulating $A$ ”. Please also see the response to reviewer #1.

Secondly, we have tested the difference of LMA across life forms using one-way ANOVA and Duncan multiple comparison. The results showed that no difference of LMA was found among life forms. Consequently, lines 309-310 have been removed. We have tested the role of leaf structure (leaf thickness (LT) and leaf density (LD)) in $A$, $g_m$ and $V_{cmax}$, and rephrased the Section "4.1 Co-variation in $g_s$, $g_m$ and $V_{cmax}$ in regulating $A$ ”. Please also see the response to reviewer #1.

*(5) Line 372: Species with low LMA may have thick cell walls in mesophyll and chloroplast.*

Response: Thank you for your suggestion. We have tested the difference of LMA across life forms using one-way ANOVA and Duncan multiple comparison. The results showed that no difference of LMA was found among life forms. Meanwhile, We have tested the role of leaf structure (leaf thickness (LT) and leaf density (LD)) in $A$, $g_m$ and $V_{cmax}$. The results showed that leaf structure plays important role in

regulating $g_m$ and $V_{cmax}$, consequently, in determining $A$. Consequently, we revised the corresponding section in "4.1 Co-variation in $g_s$, $g_m$ and $V_{cmax}$ in regulating $A$ " as "The importance of $g_m$ in constraining $A$ was variable, and depended on leaf structural traits, only LMA, LT, and LD were analyzed in this study. Large variability in $g_m$ has been shown both between and within species with different life forms and habits (Gago et al., 2014; Flexas et al., 2016). Variability in $g_m$ in this study is similar to that in global datasets (Gago et al., 2014; Flexas et al., 2016). There was no significantly difference among life forms (P>0.05). Previous studies have confirmed that LMA (Tomas et al., 2013), thickness of leaf cell wall (Peguero-Pina et al., 2017b), liquid phase of mesophyll (Veromann-Jurgenson et al., 2017), cell wall thickness of mesophyll (Terashima et al., 2011;Tosens et al., 2016), and surface area of mesophyll and chloroplast exposed to intercellular space (Veromann-Jurgenson et al., 2017) were the main limitations for $g_m$. The wide variability of $g_m$ between different species and life forms in the same ecosystem seems to be related to the diversity of leaf anatomical traits.

No significant difference of LMA, LT, and LD was found among life forms (P<0.05). The negative correlation of $g_m$ (Terashima et al., 2005) or $g_m$/LMA (Niinemets et al., 2009; Veromann-Jurgenson et al., 2017) with LMA have been reported. In this study, there was a significant relationship between $g_m$/LMA with LMA (P<0.01), however, no relationship was found between $g_m$ with LMA. $g_m$/LMA was significantly negative related to LD (p<0.01) (Fig. S5c), and weak negative related to LT (p=0.06) (Fig. S5d), demonstrating that the negative role of cell wall thickness on $g_m$ (Terashima et al., 2006; Niinemets et al., 2009). The strong investment in supportive structures was the main reason for the limitation of $g_m$ on $A$ (Veromann-Jurgenson et al., 2017). However, it is still unknown how leaf anatomical traits affect $g_m$ and $A$, and this should be further explored. "

*(6) Line 381-382: In your results, gs and gm are positively correlated, why did you conclude gm is a compensate for reductions in gs? Did you observe an increasing of gm when gs decreased.*

Response: Thank you for your comment. We corrected this mistake, and we rephrased this paragraph as: "$g_s$ is responsible for $CO_2$ exchange between atmosphere and leaf, and regulate the $CO_2$ fixation ($A$) and water loss (Lawsonand Blatt, 2014). The variability of $g_s$ was controlled by stomatal anatomy, i.e. stomata density and size, and mesophyll demands for $CO_2$ (Lawsonand Blatt, 2014). However, the stomatal anatomy was not analyzed in this study. We only focused on how the relationship between $g_s$ and $g_m$ regulate $A$. Positive relationship between $g_s$ and $g_m$ has been observed (Flexas et al., 2013). For example. the restricted $CO_2$ diffusion from the ambient air to chloroplast is the main reason for a decreased $A$ under water stress conditions due to both the stomatal and mesophyll limitations (Olsovska et al., 2016). $g_s$ was significantly positive related to $g_m$ for 63 species (P<0.001, Fig. S1) in this study, and no difference of the slopes of regression lines between $g_s$ and $g_m$ was found among life forms, demonstrating that $A$ was regulated by the co-variation of $g_s$ and $g_m$. However, the variability of $g_m$ and $l_m$ was larger than $g_s$ and $l_s$, respectively (Fig.1 and Fig.S3)."

*(7) Line 384-389: I don't think you have enough evidences to state "there was a trend of increasing lm with increasing leaf N:P", unless you add this part of research in your draft.*

Response: Thank you for your comment. There was no significant statistical relationship between $l_m$ and leaf N:P (P=0.66). We corrected this mistake, and rephrased this paragraph : " The wide variation range of $l_b$ (0.11-0.68) highlighted the importance role of $V_{cmax}$ in regulating $A$. $V_{cmax}$ was used to represent the $CO_2$ demand in photosynthetic process in this study.The relative contribution of $V_{cmax}$ to $A$ not only depends on $C_a$-$C_c$, but also on leaf nutrient levels. Positive relationship was found between $C_a$-$C_c$ and $V_{cmax}$ (Fig. 1d). And the $V_{cmax}$/LMA was co-regulated by leaf N, P and Mg content (Jing et al. 2018). In addition, $V_{cmax}$/LMA was negatively related to LT (p<0.05) (Fig. S6c) and LD (p<0.05) (Fig. S6d), while $V_{cmax}$ was not correlated to LT and LD (Fig. S6a,b), demonstrating that leaf structure plays an important role in regulating $V_{cmax}$."

*(8) Awful sentences, Lines 39-35, should split into short sentences*

Response: Rephrased as: "The results showed that (1) $g_s$ and $g_m$ varied about 7.6- and 34.5-fold, respectively, and $g_s$ was positively related to $g_m$. The contribution of $g_m$ to leaf $CO_2$ gradient was similar to that of $g_s$. The $g_s/A$, $g_m/A$ and $g_t/A$ was negative related to $V_{cmax}/A$. (2) the relative limitations of $g_s$ ($l_s$), $g_m$ ($l_m$) and $V_{cmax}$ ($l_b$) to $A$ for the whole group (combined 6 life forms) were significantly different from each other (P<0.05). $l_m$ was the largest ($0.38\pm0.12$), followed by $l_b$ ($0.34\pm0.14$) and $l_s$ ($0.28\pm0.07$). No significant difference was found between $l_s$, $l_m$, and $l_b$ for Trees and Tree/shrubs, while $l_m$ was the largest, followed by $l_b$ and $l_s$ for Shrubs, Grasses, Viens and Ferns (P<0.05). (3) iWUE varied about 3-fold (from 29.52 to 88.92 µmol $CO_2$ $mol^{-1}$ $H_2O$) across all species, and was significantly correlated with $g_s$, $V_{cmax}$, $g_m/g_s$, and $V_{cmax}/g_s$."

**Appendix**

**1 Figures**

[revised manuscript text omitted]

Figure S4 Relationship between (a) light-saturated net photosynthesis (*A*) and the leaf thickness (LT); (b) *A* and he leaf density (LD); (c) the ratio of *A* to leaf mass per area (LMA) (*A*/LMA); and (d) *A*/LMA and LD. Lines refer to regression line for 63 species. T, TS, S, G, V, and F represent Tree, Tree/Shrub, Shrub, Grass, Vine, and Fern, respectively.

[Figure]

Figure S5 Relationship between (a) the mesophyll conductance to $CO_2$ ($g_m$) and the leaf thickness (LT); (b) $g_m$ and he leaf density (LD); (c) the ratio of $g_m$ to leaf mass per area (LMA) ($g_m$/LMA); and (d) $g_m$/LMA and LD. Lines refer to regression line for 63 species. T, TS, S, G, V, and F represent Tree, Tree/Shrub, Shrub, Grass, Vine, and Fern, respectively.

[Figure]

Figure S6 Relationship between (a) the maximum carboxylase activity of Rubisco ($V_{cmax}$) and the leaf thickness (LT); (b) $V_{cmax}$ and he leaf density (LD); (c) the ratio of $V_{cmax}$ to leaf mass per area (LMA) ($V_{cmax}$/LMA); and (d) $V_{cmax}$/LMA and LD. Lines refer to regression line for 63 species. T, TS, S, G, V, and F represent Tree, Tree/Shrub, Shrub, Grass, Vine, and Fern, respectively.

**2 Tables**

Table S1 Details information about the 63 species in the subtropical primary forest in Southwest China.

| Species | Plant family | | Life form | |
|---|---|---|---|---|
| Broussonetia papyifera (Linn.) L'Hert. ex Vent. | Moraceae | Tree | Deciduous | Woody |
| Machilus microcarpa Hemsl. | Lauraceae | Tree | Evergreen | Woody |
| Melia azedarach L. | Meliaceae | Tree | Deciduous | Woody |
| Populus ×canadensis Moench. | Salicaceae | Tree | Deciduous | Woody |
| Camptotheca acuminata Decne. | Nyssaceae | Tree | Deciduous | Woody |
| Cinnamomum bodinieri Levl. | Lauraceae | Tree | Evergreen | Woody |
| Catalpa ovata G. Don | Bignoniaceae | Tree | Deciduous | Woody |
| Toona sinensis (A. Juss.) Roem. | Meliaceae | Tree | Deciduous | Woody |
| Sapium sebiferum (Linn.) Roxb. | Euphorbiaceae | Tree | Deciduous | Woody |
| Cladrastis platycarpa (Maxim.) Makino | Leguminosae | Tree | Deciduous | Woody |
| Ulmus pumila L. | Ulmaceae | Tree | Deciduous | Woody |
| Ilex macrocarpa Oliv. | Aquifoliaceae | Tree | Deciduous | Woody |
| Vitex canescens Kurz | Verbenaceae | Tree | Deciduous | Woody |
| Eriobotrya japonica (Thunb.) Lindl. | Rosaceae | Tree | Evergreen | Woody |
| Morus alba L. | Moraceae | Tree | Deciduous | Woody |
| Prunus salicina Lindl. | Rosaceae | Tree | Deciduous | Woody |
| Eucommia ulmoides Oliver | Eucommiaceae | Tree | Deciduous | Woody |
| Platycarya strobilacea Sieb. et Zucc. | Juglandaceae | Tree | Deciduous | Woody |
| Kalopanax septemlobus (Thunb.) Koidz. | Araliaceae | Tree | Deciduous | Woody |
| Zanthoxylum armatum DC. | Rutaceae | Tree | Deciduous | Woody |
| Pyrus calleryana | Rosaceae | Tree | Deciduous | Woody |
| Amygdalus persica L. var. | Rosaceae | Tree | Deciduous | Woody |

| | | | | |
|---|---|---|---|---|
| Euonymus meaackii Rupr. | Celastraceae | Tree | Deciduous | Woody |
| Zanthoxylum ovalifolium Wight | Rutaceae | Tree | Deciduous | Woody |
| Cerasus scopulorum (Koehne) Yu et Li | Rosaceae | Tree | Deciduous | Woody |
| Carpinus pubescens Burk. | Betulaceae | Tree | Deciduous | Woody |
| Lithocarpus confinis Huang | Fagaceae | Tree | Evergreen | Woody |
| Celtis sinensis Pers. | Ulmaceae | Tree | Deciduous | Woody |
| Diospyros kaki Thunb. var. silvestris Makino | Ebenaceae | Tree | Deciduous | Woody |
| Ligustrum lucidum Ait. | Oleaceae | Tree/Shrub | Deciduous | Woody |
| Rhamnus leptophylla Schneid. | Rhamnaceae | Tree/Shrub | Deciduous | Woody |
| Lindera communis Hemsl. | Lauraceae | Tree/Shrub | Evergreen | Woody |
| Itea yunnanensis Franch | Saxifragaceae | Tree/Shrub | Evergreen | Woody |
| Pittosporum brevicalyx (Oliv.) Gagnep | Pittosporaceae | Tree/Shrub | Evergreen | Woody |
| Litsea rubescens Lec. | Lauraceae | Tree/Shrub | Deciduous | Woody |
| Rhus chinensis Mill. | Anacardiaceae | Tree/Shrub | Deciduous | Woody |
| Alangium chinense (Lour.) Harms | Alangiaceae | Tree/Shrub | Deciduous | Woody |
| Evodia rutaecarpa (Juss.) Benth. | Rutaceae | Tree/Shrub | Deciduous | Woody |
| Machilus cavaleriei Levl. | Lauraceae | Tree/Shrub | Evergreen | Woody |
| Debregeasia longifolia (Burm. f.) Wedd. | Urticaceae | Tree/Shrub | Deciduous | Woody |
| Ziziphus jujuba Mill. var. spinosa (Bunge) Hu ex H. F. Chow | Rhamnaceae | Shrub | Deciduous | Woody |
| Rubus inopertus (Diels) Focke | Rosaceae | Shrub | Deciduous | Woody |
| Coriaria nepalensis Wall. | Coriariaceae | Shrub | Deciduous | Woody |
| Celastrus orbiculatus Thunb. | Celastraceae | Shrub | Deciduous | Woody |
| Wikstroemia scytophylla Diels | Thymelaeaceae | Shrub | Deciduous | Woody |
| Viburnum foetidum Wall. var. ceanothoides (C. H. Wright) Hand.-Mazz. | Caprifoliaceae | Shrub | Deciduous | Woody |
| Hedera nepalensis K. Koch var. sinensis (Tobl.) Rehd. | Araliaceae | Shrub | Deciduous | Woody |
| Rubus parvifolius L. | Rosaceae | Shrub | Deciduous | Woody |

| | | | | |
|---|---|---|---|---|
| Rosa roxbunghii | Rosaceae | Shrub | Deciduous | Woody |
| Mallotus repandus (Willd.) Muell. Arg. | Euphorbiaceae | Shrub | Deciduous | Woody |
| Mahonia bealei (Fort.) Carr. | Berberidaceae | Shrub | Evergreen | Woody |
| Fallopia multiflora (Thunb.) Harald. | Polygonaceae | Grass | | Herb |
| Conyza canadensis (L.) Cronq. | Compositae | Grass | | Herb |
| Ipomoea batatas (L.) Lam. | Convolvulaceae | Grass | | Herb |
| Senecio scandens Buch.-Ham. ex D. Don | Compositae | Grass | | Herb |
| Vitis piasezkii Maxim. | Vitaceae | Vien | Deciduous | Woody |
| Clematis urophylla Franch. | Ranunculaceae | Vien | Deciduous | Woody |
| Bauhinia glauca (Wall. ex Benth.) Benth. | Leguminosae | Vien | Evergreen | Woody |
| Caesalpinia decapetala (Roth) Alston | Leguminosae | Vien | Deciduous | Woody |
| Paederia scandens (Lour.) Merr. | Rubiaceae | Vien | | Herb |
| Cyclosorus parasiticus (L.) Farwell. | Thelypteridaceae | Fern | | |
| Cyrtomium fortunei J. Sm. | Dryopteridaceae | Fern | | |
| Pteris vittata L. | Pteridaceae | Fern | | |

Table S2 Coefficients of determination of linear regressions of fig. 1-4 and fig.6-7.

| Subgraph | Life form | Fig.1 $R^2$ | Fig.1 P | Fig.2 $R^2$ | Fig.2 P | Fig.3 $R^2$ | Fig.3 P | Fig.4 $R^2$ | Fig.4 P | Fig.6 $R^2$ | Fig.6 P | Fig.7 $R^2$ | Fig.7 P |
|---|---|---|---|---|---|---|---|---|---|---|---|---|---|
| a | Total | 0.35 | 0.000 | 0.09 | 0.018 | 0.67 | 0.000 | 0.19 | 0.000 | 0.00 | 0.922 | 0.20 | 0.000 |
| | Tree | 0.49 | 0.000 | 0.14 | 0.048 | 0.67 | 0.000 | 0.42 | 0.000 | 0.03 | 0.401 | 0.11 | 0.083 |
| | Tree/Shrub | 0.70 | 0.001 | 0.49 | 0.016 | 0.79 | 0.000 | 0.57 | 0.007 | 0.24 | 0.126 | 0.07 | 0.438 |
| | Shrub | 0.29 | 0.085 | 0.10 | 0.350 | 0.78 | 0.000 | 0.11 | 0.314 | 0.00 | 1.000 | 0.20 | 0.173 |
| b | Total | 0.75 | 0.000 | 0.47 | 0.000 | 0.53 | 0.000 | 0.65 | 0.000 | 0.34 | 0.000 | 0.52 | 0.000 |
| | Tree | 0.85 | 0.000 | 0.53 | 0.000 | 0.42 | 0.000 | 0.80 | 0.000 | 0.49 | 0.000 | 0.58 | 0.000 |
| | Tree/Shrub | 0.84 | 0.000 | 0.67 | 0.002 | 0.68 | 0.002 | 0.78 | 0.000 | 0.70 | 0.001 | 0.78 | 0.000 |
| | Shrub | 0.60 | 0.005 | 0.50 | 0.015 | 0.75 | 0.001 | 0.42 | 0.031 | 0.22 | 0.142 | 0.56 | 0.008 |
| c | Total | 0.55 | 0.000 | 0.59 | 0.000 | 0.76 | 0.000 | 0.38 | 0.000 | 0.00 | 0.934 | | |
| | Tree | 0.68 | 0.000 | 0.67 | 0.000 | 0.70 | 0.000 | 0.63 | 0.000 | 0.01 | 0.549 | | |
| | Tree/Shrub | 0.79 | 0.000 | 0.88 | 0.000 | 0.83 | 0.000 | 0.67 | 0.002 | 0.21 | 0.162 | | |
| | Shrub | 0.50 | 0.014 | 0.55 | 0.009 | 0.84 | 0.000 | 0.23 | 0.138 | 0.01 | 0.771 | | |
| d | Total | 0.25 | 0.000 | | | 0.22 | 0.000 | 0.27 | 0.000 | 0.09 | 0.016 | | |
| | Tree | 0.36 | 0.001 | | | 0.09 | 0.121 | 0.34 | 0.001 | 0.08 | 0.133 | | |
| | Tree/Shrub | 0.40 | 0.038 | | | 0.02 | 0.714 | 0.52 | 0.013 | 0.19 | 0.180 | | |
| | Shrub | 0.04 | 0.552 | | | 0.53 | 0.011 | 0.01 | 0.734 | 0.06 | 0.471 | | |

---

## Author Response (AR2)

**The point-by-point response to the reviews**

Response to reviews of manuscript "Trade-offs between water loss and carbon gain in a subtropical primary forest on Karst soils in China" bg-2018-44

Dear Editor,

We deeply appreciate you for giving us an opportunity to revise our manuscript. Here are the point-to-point responses (responses in upright Roman) to the comments (original queries in Italic). Meanwhile, we have rephrased our manuscript title as "The strategies of water-carbon regulation of plants in a subtropical primary forest on Karst soils in China".

**Response to Associate Editor comments**

Here are the point-to-point responses to the comments (original queries in Italic).

*1)Please rephrase the starting sentence by highlighting the importance of trade off between water loss and carbon gain and its implication. And the key characteristics of Karst can be briefly introduced.*

Response: Thanks a lot for your comment and suggestion. In response, we have rephrased the starting sentence of Abstract as "Coexisting plant species in a Karst ecosystem may use diversity strategies of trade off between carbon gain and water loss to adopt to the low soil nutrient and water availability conditions." (see Page 2 lines 31-33).

Meanwhile, we have rephrased the first paragraph in Section "Introduction" as "Diversity strategies of trade off between carbon gain and water loss are critical for the survival of coexisting plant species. In order to adapt to the harsh environment, coexisting plant species develop distinct patterns of strategies of carbon-water regulation (light-saturated net photosynthesis ($A$) and intrinsic water use efficiency (iWUE)) (Sullivan et al., 2017). iWUE is the ratio of $A$ to stomatal conductance to $H_2O$ ($g_{sw}$) (Moreno-Gutierrez et al., 2012). Plants with high iWUE are better able to adapt to the nutrient- and water-limited environment (Flexas et al., 2016). Due to the greater hydraulic erosion and complex underground drainage network (Nie et al., 2014; Chen et al., 2015), Karst soils cannot retain enough nutrients and water for plant growth even though precipitation is high (1000-2000 mm) (Liu et al., 2011; Fu et al., 2012; Chen et al., 2015). Understanding of the impact of $CO_2$ diffusion and maximum carboxylase activity of Rubisco ($V_{cmax}$) on $A$ and iWUE in Karst plants can provide insight into physiological strategies of water-carbon regulation of plants used in adaptation to Karst environments at the leaf scale. Until now, variability in $A$ and iWUE has been reported only in 13 co-occurring trees and 12 vines (Chen et al., 2015), and 12 co-occurring tree species (Fu et al., 2012) in two tropical Karst forests in southwestern China.". (see Page 3 lines 57-73).

*2) In addition the manuscript needs to be carefully checked for some typos.*

Response: Thanks a lot for your comment. We have carefully checked and corrected the typos.

**Response to reviewer#1**

Specific comments

*(1) I feel the explanation and justification of the chosen methodology for measuring and calculating mesophyll conductance should be in the Materials and Methods section, not in the discussion. It takes away from your actual results.*

Response: Thank you for your suggestion. This section have been moved to Section "Materials and Methods" according to your suggestion. (see Page 7 lines 182-198).

*(2) Although an "in review" article is cited in the materials and methods, I think this is not an acceptable description of methodology (line 140). This should be written out in detail as I cannot access the information from there. I would like to have more details about leaf sampling and measurements. What were the temperature and humidity chosen for the measurements? How were the leaves collected? Did you collect leaves or twigs which you then cut under water or did you collect separate leaves which you measured in the field? Did you measure fluorescence? Could you calculate your results with the Harley method as well? It is common nowadays to confirm your results with a second method as all methods have some constraints.*

Response: Thank you for your suggestions. In response, we have revised the Section "Materials and Methods" in two aspects. Firstly, we have added more details about leaf sampling and measurements in Section "Materials and Methods". Such as, we have added the method of how to collect and prepare the leaves before $CO_2$ response curves measurements. (see Page 6 lines 154-162). Meanwhile, we have described the method and conditions of $CO_2$ response curves measurements in more detail. (see page 6 lines 164-168).

Secondly, we clarified that $g_m$ was estimated by the 'curve-fitting' method in this study (see page 7 line191). As the fluorescence was not measured in this study, the Harley method cannot be used to calculate $g_m$. Details about why we choose the 'curve-fitting' method to calculate $g_m$, and the data valid confirmation have been added. (see page 7 lines 191-198).

*(3) I would also like to see more detail and justification in the statistical analysis section of the materials and methods*

Response: Thank you for your comment. In response, we have revised the Section "2.5 Statistical analysis" in two aspects. Firstly, we have moved the Section "2.4 Quantitative analysis of limitations on *A*" to Section "2.5 Statistical analysis". (see page 9 line 251 to page 10 line 265). Secondly, we have added more details about the statistical analysis in Section "2.5 Statistical analysis" (see Section "(2) Data analysis", page 10 line3 267-283). Such as, we have added the data analysis method. (see page 10 lines 268-269). We have added the bivariate linear regressions method. (see page 10 lines 269-271). We have added what method used to compare the difference of linear regressions. (see page 10 lines 273-279).

*(4) In the results, you bring out that gs was better correlated with A, but lm was more limiting. This would be important to discuss in detail in the discussion. This is an extremely important result.*

Response: Thank you for your comment and suggestion. In response, we have reanalyzed our data, and revised Section "4.1 Co-variation in $g_s$, $g_m$ and $V_{cmax}$ in regulating $A$". Firstly, we analyzed the relationships between $CO_2$ diffusion conductance ($g_s$ and $g_m$) and $V_{cmax}$, compared the relative limitations of $g_s$, $g_m$ and $V_{cmax}$ to $A$, and analyzed the relationships between the limitation factors and the corresponding relative limitations. Consequently, we have revised the corresponding results in Section "4.1 Co-variation in $g_s$, $g_m$ and $V_{cmax}$ in regulating $A$". (see page 12 line 343 to page 13 line 354). In brief, $A$ was significantly correlated with $g_s$, $g_m$, and $V_{cmax}$ (Fig.3a-c). $g_s$ was positively related to $g_m$ (Fig.S1c), while no relationship was found between the $CO_2$ diffusion conductance ($g_s$ and $g_m$) and $V_{cmax}$ (Fig. S2). $l_s$, $l_m$ and $l_b$ of 63 species varied in a large range (Fig. S3), indicating plants have a diverse strategies to co-ordinate the $CO_2$ diffusion ($g_s$ and $g_m$) and $V_{cmax}$ to maintain relative high $A$. The order of factors limitations to $A$ was $l_m > l_b > l_s$ (P<0.05) (Fig.S3). Furthermore, $l_s$, $l_m$, and $l_b$ were negatively associated with $g_s$, $g_m$, and $V_{cmax}$, respectively (Fig. 4). And the relationship was stronger for $g_m$- $l_m$ ($r^2$=0.65) than $V_{cmax}$- $l_b$ ($r^2$=0.27) and $g_s$- $l_s$ ($r^2$=0.19).

Secondly, we have discussed two possible reasons of the corresponding results in Section "4.1 Co-variation in $g_s$, $g_m$ and $V_{cmax}$ in regulating $A$". (see page 13 lines 356 -363). In brief, $g_s$ was better correlated with $A$, while the results showed that $A$ was more limited by $g_m$. That could be explained by two possible reasons. Firstly, compare to the linear relationship between $A$ and $g_s$, a nonlinear trend has been found between $A$ and $g_m$ when $g_m > 0.4$ (Fig. 3a, b). Secondly, leaf structure plays an important role in regulating $g_m$ and $V_{cmax}$, consequently, in determining $A$ (Veromann-Jurgenson et al., 2017). Negative relationships between $A$/LMA and LT ($r^2 = 0.16$, $p = 0.002$), and $A$/LMA and LT ($r^2 = 0.3$, $p < 0.001$) have been observed (Fig. S4c,d), while $A$ was not correlated to LT and LD (Fig. S4a,b).

*(5) The conclusions are a bit flat, I would like to see the paragraph rephrased so it is a bit more exciting.*

Response: Thank you for your comment. The Section "Conclusions" has been rephrased as: "This study provides information of limitations of $A$ and iWUE by $g_s$, $g_m$, and $V_{cmax}$ in 63 species across 6 life forms in the field. The results showed that plants growing in Karst CZs used a diverse strategies of carbon-water regulation, but no difference was found among life forms. The co-variation of $CO_2$ supply ($g_s$ and $g_m$) and demand ($V_{cmax}$) regulated $A$, indicating that species maintain a relatively high $A$ through co-varing their leaf anatomical structure and $V_{cmax}$. iWUE was relatively low, but ranged widely, indicating that plants used the 'profligate/opportunistic' water use strategy to maintain the survival, growth, and structure of the community. iWUE was regulated by $g_s$, $V_{cmax}$, $g_m/g_s$ and $V_{cmax}/g_s$, indicating that species with high $g_m/g_s$ or $V_{cmax}/g_s$ will have to be much more competitive to response to the ongoing rapid warming and drought in the Karst CZs." . (see page 17 line 489 to page 18 line 499).

*(6) Figure 5 needs an explanation about the whiskers: are they SEs or SDs? If they are SEs, I do not find it likely that gm was indeed the most important limiter in vies and ferns, but only grasses.*

Response: Thank you for your suggestion and comment. We clarified that whickers in Figure 5 was standard deviation. The Figure 5 legend rephrased as: "Figure 5. Limitation to light-saturated net photosynthesis ($A$) in six life forms by stomatal conductance to $CO_2$ ($l_s$), mesophyll conductance to $CO_2$ ($l_m$), and the maximum carboxylase activity of Rubisco ($l_b$). Error bars denominate standard deviation.". (see page 31 lines 788-789).

Technical comments

*(7) Line 31: grammatical error, should be "plants'"*

Response: This sentence has been deleted.

*(8) Line 38: delete first "and"*

Response: Deleted. Thank you. (see page 2 line 38).

*(9) Line 38: add "their" between "measured" and "CO2"*

Response: Change has been made. Thank you. (see page 2 line 38).

*(10)Line 38: ... calculated "the" corresponding...*

Response: Change has been made. Thank you. (see page 2 line 39).

*(11) Line 73: replace "indeed" with "however"*

Response: This change has been made. (see page 3 line 79).

*(12) Line 84: within "a" leaf.*

Response: Change has been made. Thank you. (see page 4 line 90).

*(13) Line 110: delete "The". Sentences should not be started with an article before an*

*abbreviation. This is bad style.*

Response: Deleted. Thank you. (see page 5 line 116).

*(14)Lines 125 and 126: this sentence should be in the present if the soil conditions are*

*unlikely to radically change in a short period of time.*

Response: Change has been made. (see page 5 lines 132-133).

*(15) Line 130: same comment as the previous, should be in the present if this does not*

*change rapidly.*

Response: Change has been made. (see page 5 line 137).

*(16) Line 140: You cannot use "were" if the article you are citing is still in review.*
*This is chronologically incoherent.*

Response: Thank you for your suggestion and comment. The cited article has been
accepted by "Scientific Reports". And this sentence has been rephrased as "More
details were described in Wang et al. (2018)." (see page 6 lines 171-172).

*(17) Line 148: the citation is doubles, delete one*

Response: Deleted. Thank you. (See page 7 line 177).

*(18) Line 153: delete "The"*

Response: Deleted. Thank you. (See page 7 line 191).

*(19) Line 161: no need to redefine abbreviations in each section – once is enough*

Response: Change has been made. (See page 7 line 202).

*(20) Line 166: this sentence needs to be rephrased. Stomata are not a barrier inside*
*the leaf, like this sentence seems to claim.*

Response: Thank you for your suggestion and comment. Rephrased as: "Mesophyll is the barrier for $CO_2$ inside the leaf. ". (See page 8 line 208).

*(21) Line 214: last equation was 8, this should be 9*

Response: This changed have been made. Thank you. (See page 9 line 257).

*(22)Line 253: both implies 2 variables: delete "both of"*

Response: Deleted. (See page 11 line 310).

*(23) Line 256: delete "The"*

Response: Deleted. (See page 11 line 313).

*(24) Line 257: move "respectively" to the end of the sentence*

Response: Change has been made. Thank you. (See page 11 line 314).

*(25) Line 269: delete "The"*

Response: Deleted. (See page 11 line 315).

*(26) Line 271: delete "The"*

Response: Deleted. (See page 11 line 315).

*(27) Line 272: Change to "Grasses"*

Response: Change has been made. Thank you. (See page 12 line 342).

*(28) Line 273: Change to "Accordingly, grasses"*

Response: Change has been made. (See page 12 line 323).

Response: Deleted. (See page 12 line 328).

Response: Deleted. (See page 12 line 336).

*(31) Line 295: Recent work has compared Harley, Ethier and the anatomical models finding good correlations, so I would not write largely unknown, rather "to some extent"*

Response: Rephrased as: "Thus, the accuracy of each method is to some extent unknown (Warren, 2006). " (See page 7 lines 188-189).

*(32) Line 353: this sentence should be rephrased, leads to the impression that you also did ultrastructural sampling*

Response: Thank you for your suggestion and comment. Rephrased as: "The importance of $g_m$ in constraining $A$ was variable, and depended on leaf structural traits, only LMA, LT, and LD were analyzed in this study." (See page 13 lines 365-366).

*(33) Lines 368-374: chloroplasts do not have cell walls, the sentences need to berephrased*

Response: Thank you for your suggestion and comment. This mistake has been corrected: "cell wall thickness of mesophyll". (See page 13 lines 372-373).

Response: This change has been made. (See page 15 lines 424-425).

*(35)Line 411: delete the first "in this study"*

Response: Deleted. (See page 15 line 433).

*(36)Line 415: "lose" not "loss"*

Response: Corrected. Thank you. (See page 16 line 456).

*(37) Lines 416-417 "The results ...": unnecessary sentence, delete*

Response: Deleted.

*(38)Line 422: full stop missing from the end*

Response: Added. Thank you. (See page 16 line 438).

*(39) Line 424: delete "The"*

Response: This change has been made. (See page 16 line 461).

*(40) Lines 424-425 stating with "In theory": should be in the present*

Response: This change has been made. (See page 16 line 461).

*(41) Line 433: This sentence should be in the present*

This change has been made. (See page 16 line 466).

*(42) Line 448: ...inefficiency in "the" trade-off*

Response: This change has been made. (See page 17 line 485).

*(43) Line 452: "low nutrient"*

Response: This change has been made. (See page 16 lines 451).

*(44) Line 461: iWUE is not in italic in any other place*

Response: This change has been made. (See page 17 line 489).

*(45)Line 462: ...forms in "the" field*

Response: This change has been made. (See page 17 line 490).

*(46) Line 463: ... used "a" diverse*

Response: Change has been made, thank you. (See page 17 line 491).

*(47) Line 464: ... maintain "a" relatively*

Response: This change has been made. (See page 17 line 493).

*(48) Line 465: ... used "the"*

Response: Thank you for your suggestion and comment. Chang has been made. (see page 17 line 495).

*(49)Line 483: "References"*

Response: Change has been made. (See page 18 line 513).

**Response to reviewer#2**

General comments:

*(1)    The author use "Trade-offs between water loss and carbon gain" in the title, however, the whole-text actually talk about the limitation of different components on A and iWUE.*

Response: Thanks a lot for your comment. We response to this comment from two aspects. On one hand, we have rephrased our manuscript title as "The strategies of water-carbon regulation of plants in a subtropical primary forest on Karst soils in China".

On the other hand, we have revised the Section "Discussion". Firstly, we have re-organized and revised Section "4.1 The role of $g_m$ in $CO_2$ diffusion and $V_{cmax}$", and merged it with "4.2 Co-variation in $g_s$, $g_m$ and $V_{cmax}$ in regulating $A$ ". Such as, the explanation and justification of the chosen methodology for measuring and calculating $g_m$ have been moved to Section "Materials and Methods" according to Reviewer#1's comment. (see Page 7 lines 182-198). Paragraphs "Uncertainties introduced by ignoring $g_m$." have been deleted. We have revised, corrected and re-organized the paragraph "Large variability in $g_m$", and merged it with "4.2 Co-variation in $g_s$, $g_m$ and $V_{cmax}$ in regulating $A$ ". (see page 13 lines 365-377).

Secondly, we have revised the title of Section "4.2 Co-variation in $g_s$, $g_m$ and $V_{cmax}$ in regulating $A$" as "4.1 Co-variation in $g_s$, $g_m$ and $V_{cmax}$ in regulating $A$ ". And we have re-analyzed our data, and rephrased the Section "4.1 Co-variation in $g_s$, $g_m$ and $V_{cmax}$ in regulating $A$ " to discuss about the limitation of different components on A and iWUE. (see page 13 line 339 to page 15 line 412). In brief, Karst plants have a diverse strategies to co-ordinate the $CO_2$ diffusion ($g_s$ and $g_m$) and $V_{cmax}$ to maintain relative high $A$. $A$ was regulated by the co-variation of $g_s$ and $g_m$. The strong investment in supportive structures was the main reason for the limitation of $g_m$ on $A$. The wide variation range of $l_b$ (0.11-0.68) highlighted the importance role of $V_{cmax}$ in regulating $A$. The trade-off between $CO_2$ supply ($g_s$ and $g_m$) and demand (carboxylation capacity of Rubisco) can help maintain relative high $A$.

Thirdly, we have revised the title of Section "4.3 Co-variation in $g_s$, $g_m$ and $V_{cmax}$ in regulating iWUE" as "4.2 Co-variation in $g_s$, $g_m$ and $V_{cmax}$ in regulating iWUE ". To emphasize the diverse carbon-water regulation strategies of plants in Karst CZs, and highlighted the role of trade-off between carbon gain and water loss, we have revised the Section "4.2 Co-variation in $g_s$, $g_m$ and $V_{cmax}$ in regulating iWUE". (see page 15 line 430 to page 17 line 486). In brief, coexisting plant species growing in the Karst ecosystem had a diversity water use strategies. However, Karst plants tended to lose more water to gain more carbon, i.e. Karst plants used 'profligate/opportunistic' water use strategy to adopt to the low nutrient availability and water stress conditions. iWUE was correlated to $g_s$, $V_{cmax}$, $g_m/g_s$ and $V_{cmax}/g_s$.

*(2) In the method section: The species covered wide range of functional groups, including 6 life forms. What the criteria of the species selection? Because the leaf habit (evergreen or deciduous), the shade or light-demanding behaviors also will affect the strategy of plant carbon-water regulation. For example, does fern grow in the canopy or understory, how you can put them together when analyze the data? More important, the main objective of this paper was to determine and distinguish the limitations of CO2 diffusion and Vcmax on A and iWUE in different life forms Karst forest, however, you combine all species together for most analysis, actually we donot know what's the difference between different life forms in Figs 1-4, 6,7. I Believe most land plant will behave in similar way to adapt to the environmental factor no matter where they grow, the interesting things is to what extent by different plants. For example, Based on Fig 5, we could not see any difference among the groups. So, I suggest the author should separate into 6 groups to see the differences of regression lines among groups for all the figures, and compare the difference among the life forms using proper statistical method.*

Response: Thank you for your comments and suggestions. We response to revised the manuscript from three aspects according to your comments and suggestions. Firstly, we have added our criteria of the species selection in Section "2.2 Leaf gas-exchange measurements". (see page 6 lines 145-152). In brief, the species sampled were selected according to their abundance in the study site. They are the main component of this forest. To distinguish the strategies of water-carbon regulation of plants among different life forms, those species were grouped into 6 life forms, including Tree, Tree/Shrub, Shrub, Grass, Vine, and Fern. "Tree/Shrub" is a kind of low wood plant between Tree and Shrub. Fern grow in understory. Vine climb up to the shrub canopy to get light. Meanwhile, we have added how to collect and sample leaves. (see page 6 lines 154-162). For example, Branches exposed to the sun were excised from the upper part of the crown or aboveground portion, and immediately re-cut under water to maintain xylem water continuity.

Secondly, we have re-analyzed our data either as a whole group (six life forms combined) or by individual life forms, and the difference between different life forms was tested using the standardized major axis (SMA) regression fits. The results showed that no significantly difference between life forms. Thus six life forms were grouped together to analyze the strategy of water-carbon regulation of plants in the whole text. The statistical method and results have been added in Section "2.5 Statistical analysis". (see page 10 lines 268-279).

Thirdly, all of data of six life forms were separately presented in Figure 1-4, 6,7 and Figure S1,S2, S4-S6 (See Supplement). Only the regression line for 63 species were presented in figures.

*(3) lines 139-140, because the A-Ci curve is the key data of this paper, author should describe in detail how this measurement was done rather than just cite other submitted papers. For example, you should introduce the height of your targeted individuals? how you can measure the sun-exposed leaf for canopy trees and climbing plants: : :.?did you measure in situ or cut down, if the latter, for A-Ci curve you normally need ca. 30 min, how you can avoid the effects of cutting on stomatal conductance because some species are very sensitive, do you have some information on the gs sensitivity for those species ïj§: : :..*

Response: Thank you for your suggestions. In response, we have added more details about leaf sampling and measurements in Section "Materials and Methods". Such as, we have added the method of how to sample and prepare the leaves before $CO_2$ response curves measurements. (see page 6 lines 154-162). In brief, branches exposed to the sun were excised from the upper part of the crown or above the ground, and immediately re-cut under water to maintain xylem water continuity. Back into the laboratory, branches were kept at $25^o$C for 30 min. Fully-expanded and mature leaves were induced for 30 minutes at a saturating light density. $CO_2$ response curves measurements were performed when $A$ and $g_s$ was stable. However, the height of targeted individuals did not measured.

Meanwhile, we have described the method and conditions of $CO_2$ response curves measurements in more detail. (see page 6 lines 164-172). In brief, the $CO_2$ response curves were measured with 11 $CO_2$ concentration gradients in chamber following the procedural guidelines described by Longand Bernacchi (2003). The photosynthetic photon flux density was 1500 µmol m$^{-2}$ s$^{-1}$. The leaf temperature was 25 ℃, controlled by the block temperature. The humidity in the leaf chamber was maintained at ambient condition.

Specific comments:

*(4) Line 267-269: There is no statistic tests of the differences of the results in figure 5,so it is not proper to give the statements in line 309-310. Figure 5 can't give any information that is about LMA. Please use data to demonstrate the relationship between LMA and other parameters instead of qualitative description.*

Response: Thank you for your comments and suggestions. We response to the comments and suggestions from two aspects. Firstly, we have analyzed the data of figure 5 using statistical method, and revised the corresponding Sections. Such as, we have added statistical method used to test the difference of the results in figure 5 in Section "2.5 Statistical analysis" "The difference of relative limitation of $g_s$, $g_m$ and $V_{cmax}$ to $A$ for life forms or as a whole group were performed using one-way ANOVA

and Duncan multiple comparison. The probability of significance was defined at $p < 0.05$." . (see page 10 lines 281-283).

Meanwhile, we have re-drew figure 5 and revised corresponding results in the Section "3.2 Contribution of $g_s$, $g_m$ and $V_{cmax}$ to $A$". (see page 11 line 314 to page 12 line 325 ). In brief, the contributions by $g_s$, $g_m$, and $V_{cmax}$ to limiting $A$ were different for each species (Fig. S3). Overall, $l_m$ was significantly larger than $l_b$, and $l_s$ ($P < 0.05$). There was no significantly difference between $l_s$, $l_m$ and $l_b$ for Trees and Tree/shrubs. $l_m$ of Shrubs and Grasses was significantly higher than that of $l_s$ and $l_b$ ($P < 0.05$). $l_m$ of Vines and Ferns was significantly higher than that of $l_s$ ($P < 0.05$) (Fig. 5). Meanwhile, we have revised the corresponding results and discussions in Section "4.1 Co-variation in $g_s$, $g_m$ and $V_{cmax}$ in regulating $A$ ". (see page 12 line 340 to page 13 line 354).

Secondly, we have tested the difference of LMA across life forms using one-way ANOVA and Duncan multiple comparison. The results showed that no difference of LMA was found among life forms. Consequently, lines 309-310 have been removed. We have tested the role of leaf structure (leaf thickness (LT) and leaf density (LD)) in $A$, $g_m$ and $V_{cmax}$, and rephrased the Section "4.1 Co-variation in $g_s$, $g_m$ and $V_{cmax}$ in regulating $A$ ". (see page 13 line 365 to page 14 line 389). In brief, No significant difference of LMA, LT, and LD was found among life forms ($P < 0.05$). There was a significant relationship between $g_m$/LMA with LMA ($P < 0.01$), however, no relationship was found between $g_m$ with LMA. $g_m$/LMA was significantly negative related to LD ($p < 0.01$) (Fig. S5c), and weak negative related to LT ($p = 0.06$) (Fig. S5d), demonstrating that the negative role of cell wall thickness on $g_m$ (Terashima et al., 2006; Niinemets et al., 2009).

*(5) Line 372: Species with low LMA may have thick cell walls in mesophyll and chloroplast.*

Response: Thank you for your suggestion. We have tested the difference of LMA across life forms using one-way ANOVA and Duncan multiple comparison. The results showed that no difference of LMA was found among life forms. Meanwhile, We have tested the role of leaf structure (leaf thickness (LT) and leaf density (LD)) in $A$, $g_m$ and $V_{cmax}$. The results showed that leaf structure plays important role in regulating $g_m$ and $V_{cmax}$, consequently, in determining $A$. Thus, we revised the corresponding section in "4.1 Co-variation in $g_s$, $g_m$ and $V_{cmax}$ in regulating $A$ ". (see page 12 line 340 to page 13 line 354). In brief, No significant difference of LMA, LT, and LD was found among life forms ($P<0.05$). There was a significant relationship between $g_m$/LMA with LMA ($P<0.01$), however, no relationship was found between $g_m$ with LMA. $g_m$/LMA was significantly negative related to LD ($p<0.01$) (Fig. S5c), and weak negative related to LT ($p=0.06$) (Fig. S5d), demonstrating that the negative role of cell wall thickness on $g_m$ (Terashima et al., 2006; Niinemets et al., 2009).

*(6) Line 381-382: In your results, gs and gm are positively correlated, why did you conclude gm is a compensate for reductions in gs? Did you observe an increasing of gm when gs decreased.*

Response: Thank you for your comment. We corrected this mistake, and we rephrased this paragraph as: "$g_s$ is responsible for $CO_2$ exchange between atmosphere and leaf, and regulate the $CO_2$ fixation ($A$) and water loss (Lawsonand Blatt, 2014). The variability of $g_s$ was controlled by stomatal anatomy, i.e. stomata density and size, and mesophyll demands for $CO_2$ (Lawsonand Blatt, 2014). However, the stomatal anatomy was not analyzed in this study. We only focused on how the relationship between $g_s$ and $g_m$ regulate $A$. Positive relationship between $g_s$ and $g_m$ has been observed (Flexas et al., 2013). For example. the restricted $CO_2$ diffusion from the ambient air to chloroplast is the main reason for a decreased $A$ under water stress conditions due to both the stomatal and mesophyll limitations (Olsovska et al., 2016). $g_s$ was significantly positive related to $g_m$ for 63 species ($P<0.001$, Fig. S1) in this study, and no difference of the slopes of regression lines between $g_s$ and $g_m$ was found among life forms, demonstrating that $A$ was regulated by the co-variation of $g_s$ and $g_m$. However, the variability of $g_m$ and $l_m$ was larger than $g_s$ and $l_s$, respectively (Fig.1 and Fig.S3)." (see page 14 lines 391-403).

*(7) Line 384-389: I don't think you have enough evidences to state "there was a trend of increasing lm with increasing leaf N:P", unless you add this part of research in your draft.*

Response: Thank you for your comment. There was no significant statistical relationship between $l_m$ and leaf N:P (P=0.66). We corrected this mistake, and rephrased this paragraph : " The wide variation range of $l_b$ (0.11-0.68) highlighted the importance role of $V_{cmax}$ in regulating $A$. $V_{cmax}$ was used to represent the $CO_2$ demand in photosynthetic process in this study. The relative contribution of $V_{cmax}$ to $A$ not only depends on $C_a$-$C_c$, but also on leaf nutrient levels. Positive relationship was found between $C_a$-$C_c$ and $V_{cmax}$ (Fig. 1d). And the $V_{cmax}$/LMA was co-regulated by leaf N, P and Mg content (Jing et al. 2018). In addition, $V_{cmax}$/LMA was negatively related to LT (p<0.05) (Fig. S6c) and LD (p<0.05) (Fig. S6d), while $V_{cmax}$ was not correlated to LT and LD (Fig. S6a,b), demonstrating that leaf structure plays an important role in regulating $V_{cmax}$." . (see page 14 line 405 to page 15 line 412).

*(8) Awful sentences, Lines 39-35, should split into short sentences*

Response: Rephrased as: "The results showed that $g_s$ and $g_m$ varied about 7.6- and 34.5-fold, respectively, and $g_s$ was positively related to $g_m$. The contribution of $g_m$ to leaf $CO_2$ gradient was similar to that of $g_s$. $g_s$/$A$, $g_m$/$A$ and $g_t$/$A$ was negative related to $V_{cmax}$/$A$. The relative limitations of $g_s$ ($l_s$), $g_m$ ($l_m$) and $V_{cmax}$ ($l_b$) to $A$ for the whole group (combined 6 life forms) were significantly different from each other (P<0.05). $l_m$ was the largest (0.38±0.12), followed by $l_b$ (0.34±0.14) and $l_s$ (0.28±0.07). No significant difference was found between $l_s$, $l_m$, and $l_b$ for Trees and Tree/shrubs, while $l_m$ was the largest, followed by $l_b$ and $l_s$ for Shrubs, Grasses, Viens and Ferns (P<0.05). iWUE varied about 3-fold (from 29.52 to 88.92 μmol $CO_2$ mol$^{-1}$ $H_2O$)

across all species, and was significantly correlated with $g_s$, $V_{cmax}$, $g_m/g_s$, and $V_{cmax}/g_s$." (see page 3 lines 40-50).

**The list of all relevant changes made in the manuscript**

Here are the relevant changes made in the manuscript.

(1) Page 1 lines 1-2: The title has been changed as "The strategies of water-carbon regulation of plants in a subtropical primary forest on Karst soils in China".

(2) Page 2 lines 31-33: We have rephrased the starting sentence of Abstract.

(3) Page 2 lines 40-48: The results in Abstract have been split into short sentences according to reviewer #2's comment.

(4) Page 3 lines 57-63 and line 67-70: We have rephrased the starting paragraph of Instruction to highlighting the importance of trade off between water loss and carbon gain and its implication.

(5) Page 3 lines 63-67: The key characteristics of Karst have been rephrased.

(6) Page 3 line 79: "indeed" changed to "however".

(7) Page 4 line 90: "within leaf" changed to "within a leaf".

(8) Page 4 lines 113-114: "(29 trees, 11 trees/shrubs, 11 shrubs, 4 grasses, 5 vines, and 3 ferns)" changed to "(Tree (n=29), Tree/Shrub (n=11), Shrub (n=11), Grass (n=11), Vine (n=5), and Fern (n=3))".

(9) Page 5 lines 120-121: added ", and to understanding the patterns of strategies of carbon-water regulation of Karst plants. "

(10) Page 5 lines 132, 133, 134 and 137: "was" changed to "is".

(11) Page 5 line 141: "(Wang et al., in review)" changed to " (Wang et al., 2018)".

(12) Page 6 lines 144-152: The criteria of the species selection have been added.

(13) Page 6 lines 154-172: Details of leaf sampling and measurements of the CO2 response curve have been added.

(14) Page 7 line 174: "2.3 Response curve analyses" has been added as a new section in "Materials and Methods".

(15) Page 7 lines 182-198: The explanation and justification of the chosen methodology for measuring have been moved from "Discussion" Section to "Materials and Methods" Section.

(16) Page 7 line 200: "2.3" changed to "2.4".

(17) Page 8 line 208: "Besides stomata, mesophyll is another barrier for $CO_2$ inside the leaf." changed to "Mesophyll is the barrier for $CO_2$ inside the leaf.".

(18) Page 9 line 257: "8" changed to "9".

(19) Page 9 line 259: "9" changed to "10".

(20) Page 9 line 261: "10" changed to "11".

(21) Page 10 lines 267-283: More detail and justification in the statistical analysis section have been added.

(22) Page 11 line 312: "Eqs. (8), (9) and (10)" changed to " Eqs. (9), (10) and (11)".

(23) Page 11 line 312: "contributions" changed to " limitations".

(24) Page 11 line 314: ", respectively, " changed to " , respectively.".

(25) Page 11 line 315 to page 12 line 325: we have reanalyzed our data, and revised corresponding results.

(26) Page 12 line 339: "4.2 Co-variation in gs, gm and Vcmax in regulating A " changed to "4.1 Co-variation in gs, gm and Vcmax in regulating A ".

(27) Page 12 line 340 to page 15 line 412: we have reanalyzed our data, and revised the corresponding results and discussions in Section "4.1 Co-variation in $g_s$, $g_m$ and $V_{cmax}$ in regulating $A$".

(28) Page 15 line 415: "maintain high photosynthetic efficiency " changed to "relative high A ".

(29) Page 15 line 418: "normalized gs by A (Gt =gs/A) (P<0.001) (Fig. 2c)" changed to " normalized gt by A (Gt =gt/A) (P<0.001) (Fig. 2c)".

(30) Page 15 lines 424-425: "highly-efficient " changed to "highly efficient".

(31) Page 15 lines 427-428: "$g_m$ and $A$ " changed to "$g_m$, $V_{cmax}$ and $A$ ".

(32) Page 15 line 430: "4.3 Co-variation of $g_s$, $g_m$ and $V_{cmax}$ in regulating iWUE" changed to "4.2 Co-variation of $g_s$, $g_m$ and $V_{cmax}$ in regulating iWUE".

(33) Page 17 line 489 to page 18 line 499: "Conclusions" section has been rephrased.

(34) Page 18 line 513: "6 Reference" changed to "6 References".

(35) Page 24 lines 701-706: Two reference article have been added.

(36) Page 24 lines 722-724: The reference article information has been changed.

(37) Figures 1-7 have been re-drew.

[revised manuscript text omitted]